# Learning symmetries via weight-sharing with doubly stochastic tensors

**Putri A. van der Linden**[1,*]  **Alejandro García-Castellanos**[1]  **Sharvaree Vadgama**[1]
**Thijs P. Kuipers**[2,3]  **Erik J. Bekkers**[1]

[1]Amsterdam Machine Learning Lab, University of Amsterdam
[2]Department of Biomedical Engineering and Physics, Amsterdam UMC, the Netherlands
[3]Department of Radiology and Nuclear Medicine, Amsterdam UMC, the Netherlands

## Abstract

Group equivariance has emerged as a valuable inductive bias in deep learning, enhancing generalization, data efficiency, and robustness. Classically, group equivariant methods require the groups of interest to be known beforehand, which may not be realistic for real-world data. Additionally, baking in fixed group equivariance may impose overly restrictive constraints on model architecture. This highlights the need for methods that can dynamically discover and apply symmetries as soft constraints. For neural network architectures, equivariance is commonly achieved through group transformations of a canonical weight tensor, resulting in weight sharing over a given group $G$. In this work, we propose to *learn* such a weight-sharing scheme by defining a collection of learnable doubly stochastic matrices that act as soft permutation matrices on canonical weight tensors, which can take regular group representations as a special case. This yields learnable kernel transformations that are jointly optimized with downstream tasks. We show that when the dataset exhibits strong symmetries, the permutation matrices will converge to regular group representations and our weight-sharing networks effectively become regular group convolutions. Additionally, the flexibility of the method enables it to effectively pick up on partial symmetries.

## 1 Introduction

Equivariance has emerged as a beneficial inductive bias in deep learning, enhancing performance across a variety of tasks. By constraining the function space to adhere to specific symmetries, models not only generalize better but also achieve greater parameter efficiency [8, 9]. For instance, integrating group symmetry principles into generative models has enhanced sample generation and efficient learning of data distributions, particularly benefiting areas such as vision and molecular generation [12, 5].

The most well-known and transformative models in equivariant deep learning are convolutional neural networks (CNNs) [18], which achieve translation equivariance by translating learnable kernels to every position in the input. This design ensures that the weights defining the kernels are shared across all translations, so that if the input is translated, the output features are correspondingly translated; in other words, *equivariance* is achieved through *weight sharing*. In the seminal work by Cohen and Welling [9], this concept was extended to generalize weight-sharing under any discrete group of symmetries, resulting in the group-equivariant CNN (G-CNN). G-CNNs enable G-equivariance to a broader range of symmetries, such as rotation, reflection, and scale [9, 4, 35, 29], thereby expanding the applicability of CNNs to more complex data transformations.

---

*Corresponding author: `<p.a.vanderlinden@uva.nl>`

38th Conference on Neural Information Processing Systems (NeurIPS 2024).

However, the impact of G-CNNs is closely tied to the presence of specific inductive biases in the data. When exact symmetries, such as E(3) group symmetries in molecular point cloud data, are known to exist, G-CNNs excel. Yet, for many types of data, including natural images and sequence data, these exact symmetries are not present, leading to overly constrained models that can suffer in performance [32, 24, 2, 31]. In scenarios with limited data and for certain critical downstream tasks, having appropriate inductive biases becomes even more crucial.

To avoid overly constraining models, symmetries must be chosen carefully to match those in the input data. This requires prior knowledge of these symmetries, which may not always be available. Furthermore, different symmetries at different scales can coexist, making manual determination of these symmetries highly impractical. To address this, multiple works have proposed partial or relaxed G-CNNs [24, 6, 2, 32]. Such models are initialized to be fully equivariant to some groups and learn from the data to partially break equivariance on a per-layer basis where necessary. However, these methods still require specifying which symmetries to include and can only achieve equivariance to subsets of these symmetries.

In this work, we tackle the challenge of specifying group symmetries upfront by introducing a general weight-sharing scheme. Our method can represent G-CNNs as a special case but is not limited to exact equivariance constraints, offering greater flexibility in handling various symmetries in the data. Inspired by the idea that group equivariance for finite groups can be achieved through weight-sharing patterns on a set of base weights [23], we propose learning the symmetries directly from the data on a per-layer basis, requiring no prior knowledge of the possible symmetries.

We leverage the fact that *regular* group representations act as permutations and that the expectation of random variables defined over this set of permutations is a doubly stochastic matrix [7]. This implies that regular partial group transformation can be approximated by a stack of doubly stochastic matrices which essentially act as (soft) permutation matrices. Consequently, we learn a set of doubly stochastic matrices through the Sinkhorn operator [28], resulting in weight-sharing under learnable group transformations.

We summarize our contributions as follows:

- We propose a novel weight-sharing scheme that can adapt to group actions when certain symmetry transformations are present in the data, enhancing model flexibility and performance.

- We present empirical results on image benchmarks, demonstrating the effectiveness of our approach in learning relevant weight-sharing schemes when there are clear symmetries.

- The proposed method outpaces models configured with known symmetries in environments where they are only partially present. Moreover, in the absence of predefined symmetries, it adeptly identifies effective weight-sharing patterns, matching the performance of fully flexible, non-weight-sharing models.

- We provide analyses of the learned symmetries on some controlled toy settings.

## 2   Related work

**Partial or relaxed equivariance**   Methods such as [24, 6, 2] learn partial equivariance by learning distributions over transformations, and thereby aim to learn partial or relaxed equivariances from data by sampling some group elements more often than others. [32, 33] relax equivariance by introducing learnable equivariance-breaking components. [11, 31] Relax equivariance constraints by parameterizing layers as (linear) combinations of fully flexible, non-constrained components and constrained equivariant components. Finally, several works model soft invariances through learning the amount of data augmentation in the data or model relevant for a given task, either through learned distributions on the group or (automatic) hyperparameter selection [6, 13, 22]. However, these methods require pre-specified sets of symmetry transformations and/or group structure to be known beforehand. In contrast, we aim to pick up the relevant symmetry transformations during training.

**Symmetry discovery methods**   Several methods have been developed for symmetry discovery by leveraging the infinitesimal generators of symmetry groups, as articulated through the Lie algebra [10, 36, 37, 21]. Works such as [10, 36, 21] focus on discovering symmetries associated with

general linear actions. In contrast, [37] extends these concepts to encompass non-linear symmetry transformations, offering capabilities for discovering partial symmetries within the data.

[26, 20] Learn the group structure via (irreducible) group representations. [26] Proposed to learn the Fourier transform of finite compact commutative groups and their corresponding bispectrum by learning to separate orbits on our dataset. This approach can be extended to non-commutative finite groups leveraging advanced unitary representation theory [20]. However, these methods are constrained to finite-dimensional groups and require specific orbit-predicting datasets. In contrast, our approach learns a relaxation of *regular group representations*—as opposed to irreducible representations. Moreover, our approach is not merely capable of learning symmetries, it subsequently utilizes them in a regular group-convolution-type architecture.

**Weight-sharing methods**  Previous studies have demonstrated that equivariance to finite groups can be achieved through weight-sharing schemes applied to model parameters. Notably, the works in [23], [39], and [38] provide foundational insights into this approach. In [39], weight-sharing patterns are learned by using a matrix that operates on flattened canonical weight tensors, effectively inducing weight sharing. They additionally prove that for finite groups, there are weight-sharing matrices capable of implementing the corresponding group convolution. However, their approach requires learning these patterns through meta-learning and modeling the weight-sharing matrix as an unconstrained tensor. In contrast, our method learns weight sharing directly in conjunction with the downstream task and enforces the matrix to be doubly stochastic, thereby representing soft permutations by design.

[38] Presents an approach closely aligned with ours, where a weight-sharing scheme is learned that is characterized by row-stochastic entries. Their method involves both inner- and outer-loop optimization and demonstrates the ability to uncover relevant weight-sharing patterns in straightforward scenarios. However, their approach does not support joint optimization of the canonical weights and weight-sharing pattern, and they acknowledge difficulties in extending their method to higher input dimensionalities. Unlike [38], we enforce both row and column stochasticity. Additionally, we can optimize for the sharing pattern and weight tensors jointly, and successfully apply our approach to more interesting data domains such as image processing.

In [30], group actions are integrated directly into the learning process of the downstream task. This method involves learning a set of generator matrices that operate via matrix multiplication on flattened input vectors. However, this approach constrains the operators to members of finite cyclic groups, which inherently limits their ability to represent more complex group structures. Furthermore, this restriction precludes the possibility of modeling partial equivariances, reducing the flexibility and applicability of the model to more diverse or complex scenarios.

## 3  Background

We begin by revisiting group convolutional methods in the context of image processing, followed by their relation to weight-sharing schemes. We then proceed to briefly cover the Sinkhorn operator, which is the main mechanism through which we acquire weight-sharing schemes. Some familiarity with group theory is assumed, and essential concepts will be outlined in the following.

**Groups**  We are interested in (symmetry) groups, which are algebraic constructs that consist of a set $G$ and a group product—which we denote as a juxtaposition—that satisfies certain axioms, such as the existence of an *identity* element $e \in G$ such that for all $g \in G$ we have $eg = ge = g$, *closure* such that for all $g, h \in G$ we have $gh \in G$, the existence of an *inverse* $g^{-1}$ for each $g$ such that $g^{-1}g = e$, and *associativity* such that for all $g, h, i \in G$ we have $(gh)i = g(hi)$.

**Representations**  In the context of geometric deep learning [8], it is most useful to think of groups as transformation groups, and the group structure describes how transformations relate to each other. Specifically, *group representations* $\rho : G \to GL(V)$ are concrete operators that transform elements in a vector space $V$ in a way that adheres to the group structure (they are group homomorphisms). That is, to each group element $g$, we can associate a linear transformation $\rho(g) \in GL(V)$, with $GL(V)$ the set of linear invertible transformations on vector space $V$.

**Group convolution**  Concretely, such representations can be used to define group convolutions. Consider feature maps $f : X \to \mathbb{R}^D$ over some domain on which a group action is defined, i.e., over a *G-space*. E.g., for images (signals over $X = \mathbb{R}^2$) we could consider the group $G = (\mathbb{R}^2, +)$ of translations, which acts on $X$ via $gx = x + y$, with $g = (y)$ a translation by $y \in \mathbb{R}^2$. While the group $G$ merely defines how two transformations $g, h \in G$ applied one after the other correspond to a net translation $gh \in G$, a representation $\rho$ concretely describes how data is transformed. E.g., signals $f : X \to \mathbb{R}$ can be transformed via the *left-regular representation* $[\rho(g)f](x) := f(g^{-1}x)$, which in the case of images and the translation group is given by $[\rho(g)f](x) = f(x - y)$.

In general, group convolution is defined as transforming a base kernel under every possible group action, and for every transformation taking the inner product with the underlying data via

$$\boxed{G\text{-convolution, inner product form:}} \qquad (k \star_G f)(g) = \langle \rho(g)k, f \rangle \,, \qquad (1)$$

with $\langle \cdot, \cdot \rangle$ denoting the inner product. For images, which are essentially functions over the group $G = (\mathbb{R}^2, +)$, and taking $\langle k, f \rangle := \int_X k(x)f(x)\mathrm{d}x$ the standard inner product, Eq. (1) boils down to the standard *cross-correlation* operator[2]:

$$\boxed{G\text{-convolution, integral form:}} \qquad (k \star f)(g) = \int_G k(g^{-1}h)f(h)\mathrm{d}h \qquad (2)$$

$$\boxed{\text{Standard } (\mathbb{R}^2, +) \text{ Convolution:}} \qquad (k \star f)(x) = \int_{\mathbb{R}^2} k(x' - x)f(x')\mathrm{d}x' \,. \qquad (3)$$

**Semi-direct product groups**  When equivariance to larger symmetry groups is desired, e.g. in the case of $G = SE(2)$ roto-translation equivariance for images with domain $X = \mathbb{R}^2$, a *lifting convolution* can be used to generate signals over the group $G$. In essence it is still of the form of (2), however integration is over $X$ instead of over $G$:

$$\boxed{G\text{-lifting Convolution:}} \qquad (k \star f)(g) = \int_X k(g^{-1}x)f(x)\mathrm{d}x \qquad (4)$$

$$\boxed{SE(2)\text{-lifting Convolution:}} \qquad (k \star f)(x, \mathbf{R}) = \int_{\mathbb{R}^2} k(\mathbf{R}^{-1}(x' - x))f(x')\mathrm{d}x' \,, \qquad (5)$$

with $g = (x, \mathbf{R}) \in (\mathbb{R}^2, +) \rtimes SO(2)$. The roto-translation group is an instance of a semi-direct product group (denoted with $\rtimes$) between the translation and rotation group, which has the practical benefit that a stack of rotated kernels can be precomputed [9], and the translation part efficiently be taken care of via optimized Conv2D operators. Namely via $(k \star f)(x, \mathbf{R}_i) = \texttt{Conv2D}[k_i, f]$, with $k_i := k(\mathbf{R}_i^{-1}x)$. This trick can also be applied for full group convolutions (2).

## 4  Method

Our objective is to uncover the underlying symmetries of datasets whose exact symmetries may not be known, in a manner that is both parameter-efficient and free from rigid group constraints. To achieve this, we re-examine regular representations and their critical role in generating various instantiations of the fundamental group convolution equation (1). Moving from the continuous setting to concrete instantiations, we derive weight-sharing from a finite-dimensional vector of base weights by interpreting regular representations as permutations. We further analyze the characteristics of this weight-sharing approach, proposing the use of doubly stochastic matrices. This analysis forms the foundation for developing weight-sharing layers that adaptively learn dataset symmetries.

### 4.1  Weight-sharing through permutations

In the current section, we first establish the connection between representations, weight-sharing and permutations. We then proceed to provide practical instantiations as ingredients for the proposed weight-sharing convolutional layers.

---

[2]Due to the equivalence between convolution and correlation via kernel reflection, we henceforth simply refer to operators of the type of (2) as convolution even though technically they are cross-correlations.

**Weight-sharing through learnable representations**   To achieve weight-sharing over a finite set of symmetries, we define *learnable representations* $\rho : G \to GL(V)$. Specifically, we assign a learnable transformation (permutation matrix) to each element in $G$. It is important to note that we refer to $G$ and $\rho$ as a "group" and "representation" in a loose sense, as we relax the homomorphism property and do not initially endow $G$ with a group product. Consequently, the collection of linear transformations does not form a group representation a priori. However, our proposed method is capable of modeling this structure in principle.

**Regular representations as permutations**   Consider the case of a continuous linear Lie group $G$, e.g. of rotations $SO(2)$, and a real signal $f : G \to \mathbb{R}$ over it. This signal is to be considered an *infinite dimensional vector* with continuous "indices" $g \in G$ that index the vector elements $f(g)$. To emphasize the resemblance of the regular representation to permutation matrices we write it as

$$\boxed{\text{Regular representation in integral form:}} \qquad [\rho(g)f](i) = \int_G P_g(i,h)f(h)\mathrm{d}h \,, \quad (6)$$

with $P_g(i,h) = \delta_{g^{-1}i}(h)$ a kernel that for each $g$ maps $h$ to a new "index" $i \in G$, where the Dirac delta essentially codes for the group product as $\delta_{g^{-1}h}(i)$ is non-zero only if $g^{-1}i = h \Leftrightarrow g \cdot h = i$.

The discrete counterpart of such an integral transform is matrix-vector multiplication with a matrix $\rho(g) = \mathbf{P}_g$ with entries $P_{ghi} = 1$ if $gh = i$ and zero otherwise. As a concrete example, consider a signal $f : G \to \mathbb{R}$ over the discrete group $G = C_4$ of cyclic permutations of size 4, i.e., a periodic signal of length four, then we could vectorize it as $f \to \mathbf{v}$ with entries $v_g = f(g)$ and the regular representation becomes a simple cyclic permutation matrix with

$$\mathbf{P}_0 = \begin{pmatrix} 1 & 0 & 0 & 0 \\ 0 & 1 & 0 & 0 \\ 0 & 0 & 1 & 0 \\ 0 & 0 & 0 & 1 \end{pmatrix}, \quad \mathbf{P}_1 = \begin{pmatrix} 0 & 0 & 0 & 1 \\ 1 & 0 & 0 & 0 \\ 0 & 1 & 0 & 0 \\ 0 & 0 & 1 & 0 \end{pmatrix}, \quad \mathbf{P}_2 = \begin{pmatrix} 0 & 0 & 1 & 0 \\ 0 & 0 & 0 & 1 \\ 1 & 0 & 0 & 0 \\ 0 & 1 & 0 & 0 \end{pmatrix}, \quad \mathbf{P}_3 = \begin{pmatrix} 0 & 1 & 0 & 0 \\ 0 & 0 & 1 & 0 \\ 0 & 0 & 0 & 1 \\ 1 & 0 & 0 & 0 \end{pmatrix}.$$

We refer to the collection of permutation matrices $\mathbf{P} \in [0,1]^{|G| \times |G| \times |G|}$ as the **permutation tensor**.

Recall that the regular convolution operator (2) is not only defined for signals over groups $G$ but for $G$-spaces $X$, in general, as in Eq. (4). It requires a representation that acts on signals (convolution kernels) over $X$. However, since the group $G$ acts by automorphisms (bijections) on the space $X$, we can define the regular representation—as before—using permutation integral with $P_g(x,x') = \delta_{g^{-1}x'}$ that effectively sends old "indices" $x'$ to their new location $x$, or concretely via permutation matrices $\mathbf{P}$ of shape $|G| \times |X| \times |X|$, where $X$ is the domain of the signal that is transformed—which can also be $G$.

In practice, it is common that we perform a discretization of continuous signals $f$, e.g., when we work with images, we discretize the continuous signal $f : \mathbb{R}^2 \to \mathbb{R}^C$ to $\mathbf{f} : \mathbb{Z}^2 \to \mathbb{R}^C$. Therefore, the group action over the discretized signal should approximate the group action over the continuous signal. We can see that in some cases, the group representation of the action on the discrete signal can still be implemented using permutation matrices. E.g., $90°$ rotations (in $C_4$) applied to images merely permute the pixels. However, for finer discretizations, e.g. using $45°$ rotations, interpolation can be used as a form of approximate permutations [17, Fig. 2].

**Weight sharing**   In a discrete group setting, we then see that convolution (1) is obtained by multiplications with matrices obtained by stacking permuted base kernel weights $\mathbf{w} \in \mathbb{R}^{|X|}$

$$\boxed{\text{Discrete } G\text{-convolution:}} \qquad \mathbf{f}^{out} = \mathbf{W}\mathbf{f}^{in} \,, \quad \text{with} \quad \mathbf{W} = \mathbf{P}\mathbf{w} := \begin{pmatrix} (\mathbf{P}_0\mathbf{w})^T \\ (\mathbf{P}_1\mathbf{w})^T \\ \vdots \end{pmatrix} \in \mathbb{R}^{|G| \times |X|} \,.$$

$$(7)$$

E.g., a group convolution over $C_4$ is implemented with a matrix of the form $\mathbf{W} = \begin{pmatrix} w_0 & w_1 & w_2 & w_3 \\ w_3 & w_0 & w_1 & w_2 \\ w_2 & w_3 & w_0 & w_1 \\ w_1 & w_2 & w_3 & w_4 \end{pmatrix}$.

**Regular representations allow for element-wise activations**   An important practical element of using regular representations to define group convolutions is that permutations commute with element-wise activations, namely, $[\rho(g)\sigma(f)](i) = \sigma(f)(g^{-1}i) = \sigma(f(g^{-1}i)) = \sigma([\rho(g)f](i))$. In contrast, steerable methods—based on irreducible representations (cf. App. A.1)—require specialized

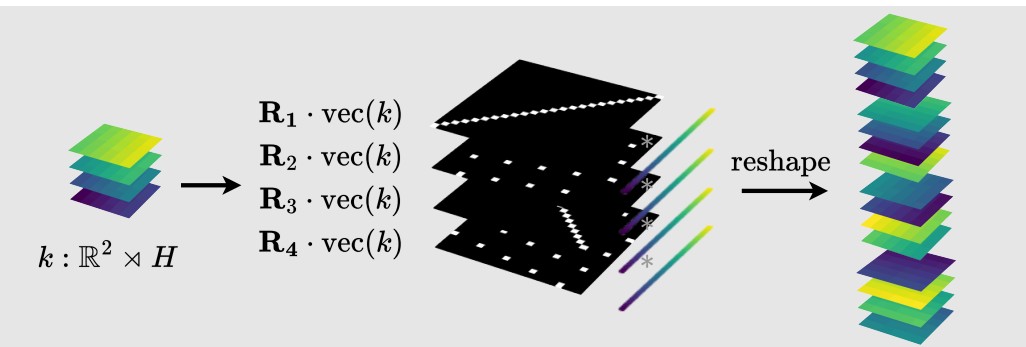

Figure 1: Kernel stacks are acquired through a learned weight-sharing scheme applied to a set of flattened base kernels.

activation functions so as not to break group equivariance. Such activations in practice are not as effective as the classic element-wise activations such as ReLU [34]: they may introduce discretization artifacts that disrupt exact equivariance, ultimately constraining the method's expressivity [5]. Hence, when learning weight-sharing schemes—as is our objective—it is preferred to achieve weight-sharing using regular representations without the risk of breaking equivariance by using standard activation functions.

## 4.2 Learnable doubly stochastic tensors

Having established the link between regular representations and soft permutations, we now motivate the use of doubly stochastic matrices as natural candidates for their implementation. Specifically, we utilize the fact that the expected value of random variables over this set of permutations yields a doubly stochastic matrix [7].

**Doubly stochastic matrices**   Let $\mathcal{S}_\infty$ $(\mathcal{S}_n)$ denote respectively the system of infinite $(n \times n)$ doubly stochastic matrices, i.e. matrices $S \equiv \{s_{ij} \in [0,1] : i,j = 1,2,...(,n)\}$ such that $\sum_j s_{ij} = 1$, and $\sum_i s_{ij} = 1$. Let $\mathcal{P}_\infty$ $(\mathcal{P}_n)$ denote respectively the system of infinite $(n \times n)$ permutation matrices, i.e. matrices $P \equiv \{p_{ij} \in \{0,1\} : i,j = 1,2,...(,n)\}$ such that $\sum_j p_{ij} = 1$, and $\sum_i p_{ij} = 1$. Note that for any permutation tensor $\mathbf{P} \in \mathbb{R}^{|G| \times |X| \times |X|}$, where $X$ is the domain of the signal that is transformed by the group $G$, then $\mathbf{P}_g \in \mathcal{P}_{|X|}$ for every $g \in G$.

Then, by Birkhoff's Theorem [7], and its extension to infinite dimensional matrices, commonly called Birkhoff's Problem 111 [14, 25, 15, 3], we have that any convex combination of permutation matrices will be equal to a doubly stochastic matrix, i.e,

$$\sum_{P \in \mathcal{P}_n} \lambda(P) P = S \in \mathcal{S}_n, \text{ with } \sum_{P \in \mathcal{P}_n} \lambda(P) = 1 \quad (\forall n \in \mathbb{N} \cup \{+\infty\})$$

where $\lambda(P)$ gives a probability measure supported on a finite subset of the set of permutation matrices $\mathcal{P}$. Therefore we may state that

Using doubly stochastic matrices, we can model approximate equivariance as defined in [24].

I.e., let $S$ be a random variable over $\{\mathbf{P}_g \in \mathcal{P}_{|X|} \mid g \in G\}$ with a finitely supported probability measure $\mathbb{P}[S = \mathbf{P}_g] = \lambda(\mathbf{P}_g)$ for every $g \in G$, then $\mathbf{S} = \mathbb{E}[S] = \sum_{g \in G} \mathbb{P}[S = \mathbf{P}_g]\mathbf{P}_g$ is a doubly stochastic matrix. We want to note that $\mathbf{S}$ can be seen as a generalization of the *convolution matrix* presented in [19].

**Sinkhorn operator**   The Sinkhorn operator [28, 1] transforms an arbitrary matrix to a doubly stochastic one through iterative row and column normalization, provided that the number of iterations

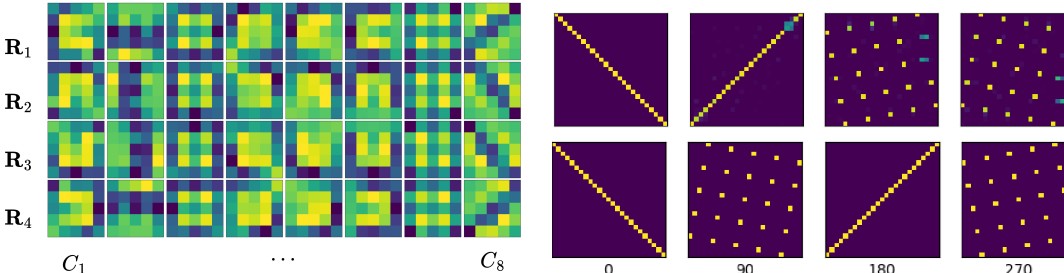

Figure 2: Learned kernels from the lifting layer of WSCNN, applied to rotated MNIST and reshaped to [No.elem., $C_{out}$]. Since $\mathbf{R}_1$ is set as the identity operator, the first column displays the raw kernels.

Figure 3: Comparison of $C_4$ representations and the representation stack learned by the lifting layer on the rotated MNIST dataset. Top: learned representations. Bottom: permutations for $C_4$ on $d = 25$.

is large enough. That is, initialize a tensor $\mathbf{X} \in \mathbb{R}^{N \times N}$, then it will converge to a doubly stochastic tensor via the following algorithm:

$$S^0(\mathbf{X}) = \exp(\mathbf{X}), \qquad S^l(\mathbf{X}) = T_c(T_r(S^{l-1}(\mathbf{X}))), \qquad \mathcal{S}_N \ni \mathbf{S} = \lim_{l \to \infty} S^l(\mathbf{X}), \quad (8)$$

with $T_c$ and $T_r$ the normalization operators over the rows and columns, respectively, defined as $T_c = X \oslash \underbrace{\mathbf{1}_N \mathbf{1}_N^T \mathbf{X}}_{\mathrm{sum}_c(\mathbf{X})}$ and $T_r = \mathbf{X} \oslash \underbrace{\mathbf{X} \mathbf{1}_N \mathbf{1}_N^T}_{\mathrm{sum}_r(\mathbf{X})}$, where $\oslash$ denotes elementwise division, $\mathrm{sum}_c(\cdot)$, $\mathrm{sum}_r(\cdot)$ perform column-wise and row-wise summation, respectively.

**Our proposal: Weight Sharing Convolutional Neural Networks**  Having established the foundational elements, we now define our *Weight Sharing Convolutional Neural Networks* (WSCNNs). We let $\Theta_i^l \in \mathbb{R}^{|X| \times |X|}$ be a collection of learnable parameters that parametrize the *representation stack* of the layer $l$ as $\mathbf{R}^l = (S^K(\Theta_0^l), S^K(\Theta_1^l), ..., S^K(\Theta_N^l))^T \in [0,1]^{|G| \times |X| \times |X|}$. i.e., we parameterize this tensor as a stack of $|G|$ approximate doubly stochastic $|X|$-dimensional matrices, wherein stochasticity is enforced via $K$ applications of the Sinkhorn operator. We also define a set of *learnable base weights* $\boldsymbol{\theta}^l \in \mathbb{R}^{|X| \times C_{out} \times C_{in}}$. The WSCNN layer is then simply given by (7) with $\mathbf{P}$ and $\mathbf{w}$ respectively replaced by $\mathbf{R}^l$ and $\boldsymbol{\theta}^l$.

We further note that on image data $|X|$ can be large, making the discrete matrix form implementation computationally demanding. Hence, we consider semi-direct product group parametrizations for $G$, in which we let $G$ be of the form $(\mathbb{R}^n, +) \rtimes H$, with $H$ a learnable (approximate) group. Then, the representation stacks will merely be of shape $|H| \times |X'| \times |X'|$, with $|H| \ll |G|$ the size of the sub-group and $|X'|$ the number of pixels that support the convolution kernel. A WSCNN layer is then efficiently implemented via a `Conv2D`$[\mathbf{f}, \mathbf{R}^l \boldsymbol{\theta}^l]$. For group convolutions (after the lifting layer) the representation stacks will be of shape $|H| \times (|X'| \times |H|) \times (|X'| \times |H|)$. Computational scaling requirements can be found in Appendix C.4.

## 5 Experiments

We first demonstrate that the proposed weight sharing method can effectively pick up on useful weight sharing patterns when trained on image datasets with different equivariance priors. We then proceed to show the method can effectively handle settings where partial symmetries are present in the data, and further analyze the learned weight-sharing structures on a suite of toy datasets. Model architectures, regularizers (norm, ent) and design choices can be found in Appendix C.4 and C.1, respectively. An analysis of computational requirements can be found in Appendix C.4

### 5.1 Image datasets and equivariance priors

We assess the efficacy of our proposed weight-sharing scheme in recognizing data symmetries through experiments on datasets subjected to various data augmentations. Specifically, we evaluate our model on MNIST images that have been rotated (with full $SO(2)$ rotations) and scaled (with scaling

Table 1: Test accuracy on MNIST for both rotation and scaling transformations. Additional parameters induced by weight-sharing are marked (+). Parameter counts denoted in millions (M) or thousands (K). Best-performing models (equivalent within $< 1\%$) marked **bold**.

| Model | # Params | Sharing Scheme | Accuracy | |
|---|---|---|---|---|
| | | | Rotations | Scaling |
| CNN | 412 K | $Z_2$ | $\mathbf{98.48 \pm .08}$ | $\mathbf{99.30 \pm .01}$ |
| GCNN | 103 K | $Z_2 \rtimes C_4$ | $\mathbf{98.96 \pm .14}$ | $97.50 \pm .15$ |
| WSCNN + norm | 410 K (+ 122 K) | Learned | $97.56 \pm .07$ | $\mathbf{99.27 \pm .04}$ |
| WSCNN + norm + ent | 410 K (+ 122 K) | Learned | $98.04 \pm .11$ | $\mathbf{99.24 \pm .01}$ |

Table 2: Test accuracy on CIFAR-10. Number of elements denotes the number of group elements used in group convolutional models. Additional parameters induced by weight-sharing are marked (+). Parameter counts denoted in millions (M) or thousands (K). Best-performing models (equivalent within $< 1\%$) marked **bold**.

| Model | # Params | # Elements | Accuracy |
|---|---|---|---|
| CNN-32 | 428 K | - | $70.50 \pm 0.62$ |
| CNN-64 | 1.66 M | - | $76.29 \pm 0.57$ |
| CNN-128 | 6.5 M | - | $\mathbf{78.83 \pm 0.01}$ |
| GCNN | 1.63 M | 4 | $76.72 \pm 0.26$ |
| WSCNN + norm | 1.63 M (+ 468 K) | 4 | $\mathbf{78.80 \pm 0.46}$ |
| WSCNN + norm + ent | 1.63 M (+ 468 K) | 4 | $76.80 \pm 1.40$ |

factors between $[0.3, 1.0]$). We categorize these datasets based on their data symmetry characteristics: MNIST with rotation and scaling as datasets with known symmetries, and CIFAR-10 with flips as a dataset with unknown symmetries.

For RotatedMNIST, we regard a C4-group convolutional model as a benchmark since it has been equipped with a subgroup of the underlying data symmetries a priori. Additionally, we contrast our results with a non-constrained CNN model which has double the number of channels, allowing for free optimization without symmetry constraints. As such, our evaluations are benchmarked against two distinct models: 1) a group convolutional model that is equivariant to discrete rotations, embodying fixed equivariance constraints, and 2) a standard CNN that adheres only to translational equivariance, without additional constraints.

This experimental setup positions the standard CNN as the most flexible model lacking predefined inductive biases. In contrast, the group convolutional neural network (GCNN) is characterized by fixed weight sharing, while our proposed weight-sharing CNN (WSCNN) introduces a semi-flexible, learnable weight-sharing mechanism. Note that we explicitly distinguish between the number of free model and kernel parameters and the additional parameters introduced by our weight sharing scheme throughout results (marked by +).

Results can be found in Tab. 1. When there is a clear misalignment between the model and data symmetries, the constraints imposed by the model hinder performance, as demonstrated by the C4-GCNN on the scaled MNIST dataset. Notably, our proposed method consistently achieves high performance across all datasets without requiring fixed group specifications. Furthermore, visual inspection of the learned kernels indicates that WSCNN adapts to the underlying data symmetries by effectively rotating kernels, as shown in Figure 2. Additionally, analysis of the learned representation stack reveals that it closely resembles elements of $C_4$ permutations, further demonstrating the model's capability to internalize and replicate data transformations (see Figure 3 and Appendix B.3).

Additionally, we test the model on CIFAR-10, representing a dataset with possibly more complex/unknown symmetry structures, and CIFAR-10 with horizontal flips (which cannot be represented by $C_N$ transformations). Results can be found in Tab. 2, with detailed training and model specifications available in Appendix C.4. We compare against the (possibly misspecified) C4-GCNN, and several unconstrained CNNs models with varying number of channels: 1) **CNN-32** matched in free

Table 3: Test accuracy on MNIST with partial rotations.

| Model | Rot. Range | Accuracy |
|---|---|---|
| GCNN | $[0, \ 90°]$ | $98.84 \pm .002$ |
| | $[0, 180°]$ | $98.72 \pm .001$ |
| WSCNN | $[0, \ 90°]$ | $98.87 \pm .001$ |
| | $[0, 180°]$ | $99.25 \pm .001$ |

Table 4: Test accuracy on CIFAR-10 with horizontal flips.

| Model | # Params | # Elements | Accuracy |
|---|---|---|---|
| CNN-64 | 1.7 M | - | $79.81 \pm .001$ |
| CNN-128 | 6.5 M | - | $82.60 \pm .001$ |
| GCNN | 1.6 M | 4 | $76.05 \pm .004$ |
| WSCNN | 1.6 M (+ 468 K) | 4 | $82.38 \pm .003$ |

kernel size, 2) **CNN-64** matched in parameter budget, and 3) **CNN-128** matched in effective kernel size (calculated as $|G| \times$ channels $= 4 \times 32 = 128$). Despite the kernel constraints in WSCNN, it achieves performance comparable to that of the unconstrained 128-channel CNN (within $< 1\%$ accuracy), at a significantly smaller parameter budget (2.1 M vs. 6.5 M).

## 5.2 Learning partial equivariances

We show our method is able to pick up on partial symmetries by testing it on MNIST with rotations sampled from a subset of SO(2) and compare it to the C4-GCNN. Additionally, we show results on CIFAR-10 with horizontal flips, which is a commonly used train augmentation. Results can be found in Tab. 3 and Tab. 4.

Additionally, We proceed to test the model's capability to detect data symmetries by applying it to a suite of toy problems, wherein the datasets comprise noisy $G$-transformed samples. Details on the data generation framework are provided in Appendix B. Our testing employs a single-layer setup aimed at learning a collection of kernels that ideally match each data sample, considering inherent noise. This involves training the model to identify a set of base kernels and their pertinent transformations, effectively adapting to the variations presented by the toy problems.

## 6 Discussion and Future Work

We demonstrated a method that can effectively identifies underlying symmetries in data, even without strict group constraints. Our approach is uniquely capable of learning both partial and approximate symmetries, which more closely mirrors the complexity found in real-world datasets. Utilizing doubly stochastic matrices to adapt kernel weights for convolutions, our method offers a flexible means of learning representation stacks, accommodating both known and unknown structures within the data. This adaptability makes it possible to detect useful patterns, although these may not always be interpretable in traditional group-theoretic terms due to the absence of predefined structures in the representation stack.

Limitations include computational requirements, which scale quadratically with the size of the group and the kernel size, posing challenges in scenarios with large groups or high-resolution data. As such, in this work we have designed the representation stack to be uniform across the channel dimension. However, this prevents learning of other commonly used image transformations such as color jitter. Furthermore, the need for task-specific regularization to manage entropy scaling during the learning of representations introduces complexity in hyperparameter tuning, which can be a barrier in some applications. Additionally, we observed that representations in later layers may show minimal diversity, suggesting that further innovation in regularization strategies might be necessary to enhance the distinctiveness of learned features across different layers.

For future work, we aim to enhance our method by implementing hierarchical weight-sharing across layers and promoting group equivariance more systematically. One promising direction is to leverage the concept of a Cayley tensor, akin to [20], to identify and reuse learned group structures across different layers of the network. This approach would not only impose a more unified and coherent group structure within the model but also potentially reduce the computational overhead associated with learning separate representations for each layer. By encouraging a shared group structure throughout the network, we anticipate improvements in both performance and interpretability, paving the way for more robust and efficient symmetry-aware learning systems.

## Acknowledgements

SV and AGC are supported by the Hybrid Intelligence Center, a 10-year program funded by the Dutch Ministry of Education, Culture and Science through the Netherlands Organisation for Scientific Research. The computational results presented have been achieved in part using the Snellius cluster at SURFsara. Additionally, we thank Alexander Timans for helpful discussions.

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

# A  Preliminaries

## A.1  Irreducible representations

In this section, we closely follow the mathematical preliminaries outlined in [34]. For a comprehensive reference on Representation Theory, see [27].

**Equivalent representations**   Two representations $\rho$ and $\rho'$ of a group $G$ are considered *equivalent* if there exists a similarity transform such that:

$$\forall g \in G \quad \rho(g) = Q\rho'(g)Q^{-1}$$

where $Q$ represents a change of basis matrix.

**Irreps**   A matrix representation is called *reducible* if it can be decomposed as:

$$\rho(g) = Q^{-1}(\rho_1(g) \oplus \rho_2(g))Q^{-1} = Q \begin{pmatrix} \rho_1(g) & 0 \\ 0 & \rho_2(g) \end{pmatrix} Q^{-1}$$

where $Q$ is a change of basis matrix. If the sub-representations $\rho_1$ and $\rho_2$ cannot be further decomposed, they are termed *irreducible representations* (*irreps*). The set of all irreducible representations of a group $G$ is denoted as $\hat{G}$.

Additionally, any representation $\rho : G \to GL(V)$ of a compact group $G$ can be expressed as:

$$\rho(G) = Q \left[ \bigoplus_{j \in \mathcal{I}} \rho_j \right] Q^{-1}$$

where $\mathcal{I}$ is an index set (possibly with repetitions) over $\hat{G}$.

Similarly to what is showed in Section 4, a representation $\rho : G \to \mathbb{R}^{d \times d}$ can be viewed as a collection of $d^2$ functions over $G$. The **Peter-Weyl theorem** asserts that the collection of functions formed by the matrix entries of all irreps in $\hat{G}$ spans the space of all square-integrable functions over $G$. For most groups, these entries form an orthogonal basis, allowing any function $f : G \to \mathbb{R}$ to be written as:

$$f(g) = \sum_{\rho_j \in \hat{G}} \sum_{m,n < d_j} w_{j,m,n} \cdot \sqrt{d_j} [\rho_j(g)]_{mn}$$

where $d_j$ is the dimension of the irrep $\rho_j$, while $m, n$ index the entries of $\rho_j$. Note that this expression corresponds to the *inverse Fourier transform* and that the coefficients $w_{j,m,n}$ can be obtained by the *Fourier transform* of $f$ with respect to the basis functions $\{[\rho_j(g)]_{mn}\}_{j \in \mathcal{I}}$.

**Connection with regular representations**   It can be shown that the regular representations can be decomposed using the corresponding irreps as follows:

$$\rho_{\text{reg}}(g) = Q^{-1} \left[ \bigoplus_{p_j} \bigoplus^{d_j} \rho_j \right] Q$$

where $Q$ performs the Fourier transform, while $Q^{-1}$ performs the inverse Fourier transform. This implies that when functions $f : G \to \mathbb{R}$ are considered as vectors in $\mathbb{R}^{|G|}$, with a basis where each axis corresponds to a group element, then, as we have seen in Section 4, the group action results in a permutation of these axes. However, applying the Fourier transform changes the basis so that $G$ acts independently on different subsets of the axes, resulting in the action being represented by a block-diagonal matrix, which is the direct sum of irreps.

# B  Toy problems

## B.1  Data generation processes

For the construction of the toy problems, we look at two types of data-generating processes:

**Equivariant data.** Assume a canonical vector $\hat{\mathbf{x}}$ and corresponding label $\hat{\mathbf{y}}$. We assume these vectors transform under a known group $G$, such that the data-generating process is as follows:

$$\text{Sample group action:} \quad g \sim \mu(G),$$
$$\text{Apply group action to feature with noise:} \quad \mathbf{x} = \rho^x(g)\hat{\mathbf{x}} + \epsilon$$
$$\text{Apply group action to label:} \quad \mathbf{y} = \rho^y(g)\hat{\mathbf{y}}$$

with $\rho^x$, $\rho^y$ the representations of $G$ acting on the feature and label space, respectively.

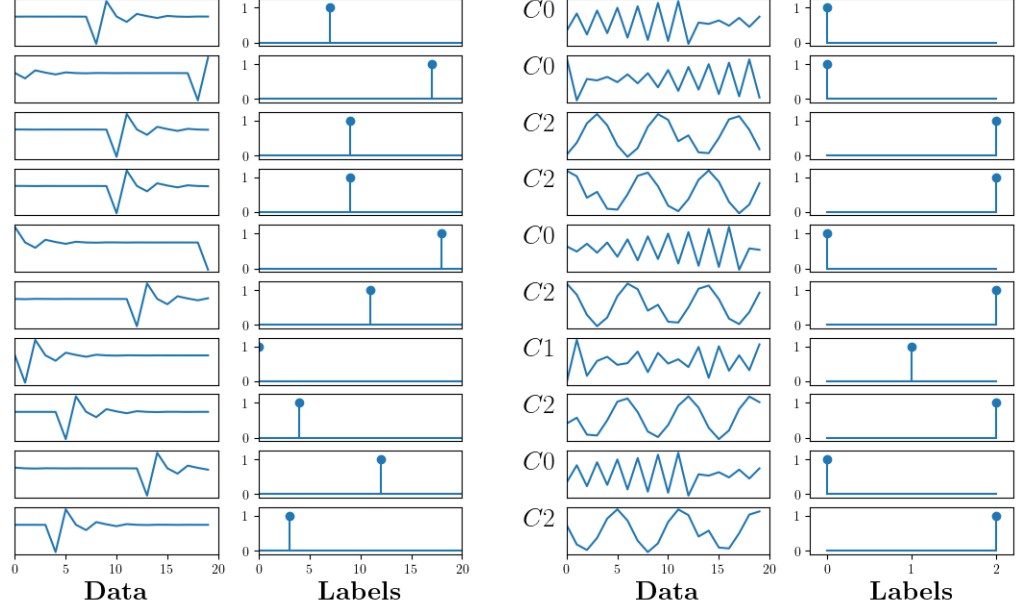

(a) Equivariant task. The labels transform with the data.     (b) Invariant task. The labels are fixed.

Figure 4: Samples of the two tasks.

**Evaluation metrics** To assess whether our method effectively captures the underlying data symmetries, we analyze the learned weight-sharing structure by comparing it to known ground truth patterns. Since our model does not impose associativity or any strict group constraints on the representation stack, it may learn to represent mixtures or interpolations of group elements. Hence, in general, we will not observe a relevant algebraic structure if we produce a Cayley table based on the learned representations as done in [26, 20].

Therefore, we will use an approach that allows us to capture the flexibility of our representations. To this end, we will examine each representation matrix to determine how closely it resembles a *convolution matrix* [19] associated with a random variable defined over some specific group $G$. We employ the set of group actions from $G$, represented as doubly stochastic tensors $\{\mathbf{P}_k^{gt}\}_{k=1}^{|G|}$, as a reference framework to quantify the fit and alignment of our model's representations with these predefined group actions. As such, for each learned weight sharing tensor $\mathbf{R}_i^l$, we calculate the fit $\hat{\mathbf{P}}_i = \sum_k^{|G|} c_k \mathbf{P}_k^{gt}$ and acquire coefficients $c_k > 0$, such that $\sum_k^{|G|} c_k = 1$ in a constrained linear regression setup by minimizing $||\hat{\mathbf{P}}_i - \mathbf{R}_i^l||_2$.

### B.2 Additional results: toy problems

We conducted experiments on various signals subjected to different transformations, including: a 1D signal with cyclic shifts (exemplary samples shown in Fig. 4a), a 2D signal with $C_8$ rotations (illustrated in Fig. 7), and a 3D voxel grid enhanced by 24 cube symmetries. In each scenario, the learned kernel stack accurately matched the data samples, achieving perfect accuracy. Figure 6

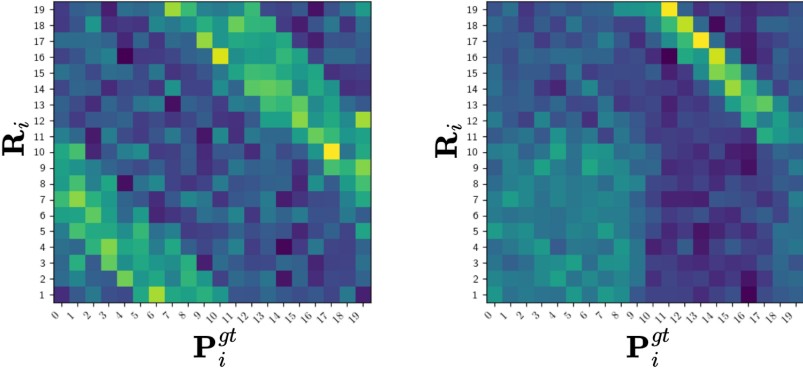

(a) Shift-dataset with uniformly sampled group elements.

(b) Shift-dataset with partially sampled group elements.

Figure 5: Coefficient responses of learned representations and their base transformations.

displays the learned representations for localized shifts in the 1D signal, while Figure 8 presents the learned kernel stack for the 2D signal dataset.

Furthermore, to assess whether our model can identify partial group structures, we evaluate it using two distinct datasets: a 1D signal enhanced with cyclical shifts, and a $3 \times 3 \times 3$ voxel grid subjected to rotations from $C4 \times C4$. For the shift dataset, we utilize the complete set of cyclical shifts as the ground truth representations. Figure 5(a, b) displays the coefficients for the shift dataset. Specifically, part (a) shows coefficients when trained uniformly across group elements, while part (b) illustrates coefficients using only the first half of the group elements, where the group no longer retains cyclic properties or satisfies closure. Given that our method does not assume cyclic groups or group closure, it effectively captures the relevant group transformations even for partial transformations. App. B.4 shows the coefficients for the base representations of $C_4 \times C_4 \times C_4$ cube symmetries. Since the data augmentation only applied $C_4 \times C_4$ transformations, the method predominantly identifies elements corresponding to these transformations, as highlighted by the red line.

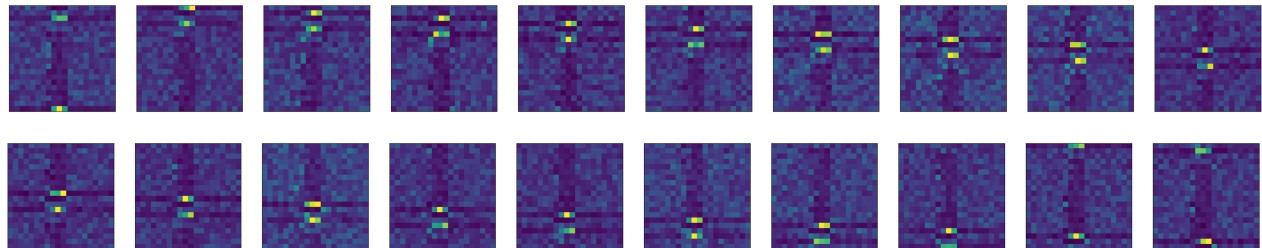

Figure 6: Representations learned for 1D equivariant shift task

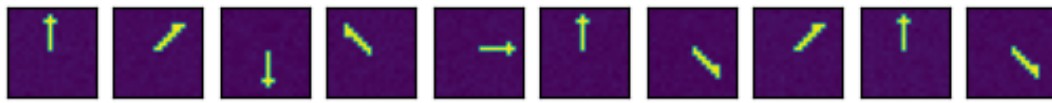

Figure 7: Samples of the 2D-signal dataset.

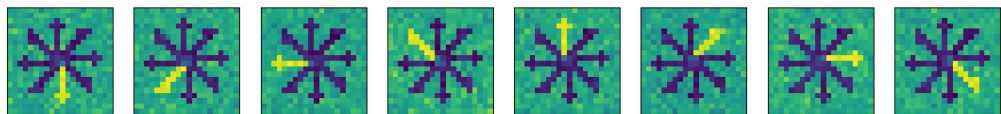

Figure 8: Equivariant task for flattened rotated 2D signals. Learned kernel stack

## B.3 Visualization of G-Conv layers

Figure 9 displays the ground truth permutation matrices that implement a shift-twist operator, which is the group transformation that underlies regular group convolution operator for $C4$ rotations. Figure 10 illustrates the corresponding matrices learned for each learnable weight sharing layer on the rotated MNIST dataset. The learned matrices closely show similar patterns as the shift-twist operator, suggesting the model's ability to capture such transformations from training data.



Figure 9: Ground-truth permutation matrices for $C_4$ rotations for $5 \times 5$ spatial kernel, implementing a *shift-twist* operator.

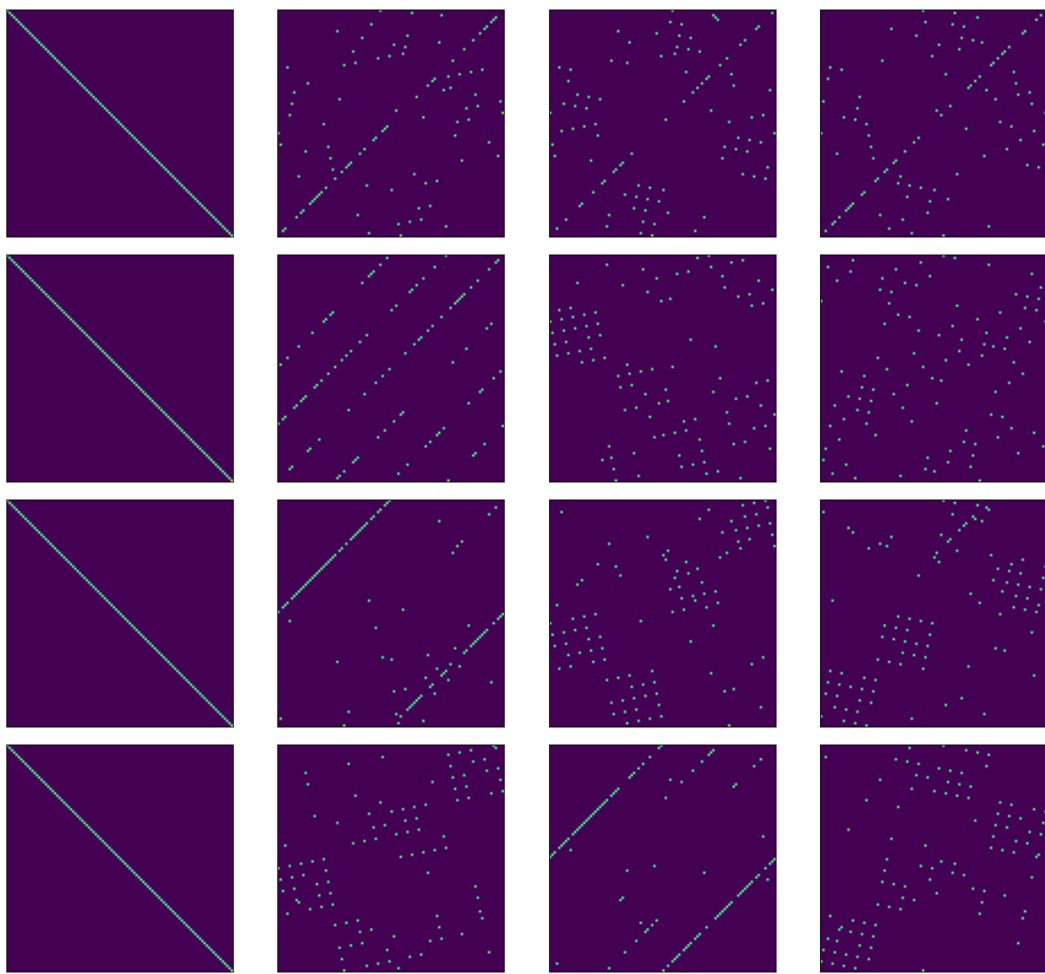

Figure 10: Learned representations for the weight-sharing G-conv layers. Top to bottom: first to last layer learned representation stacks.

## B.4  Additional results: learning partial equivariance

Fig. 11 shows the coefficients for the base representations of $C_4 \times C_4 \times C_4$ cube symmetries. The x-axis quantifies the permutation representation for each element consisting of the number of 90-degree flips around each x, y or z axis.

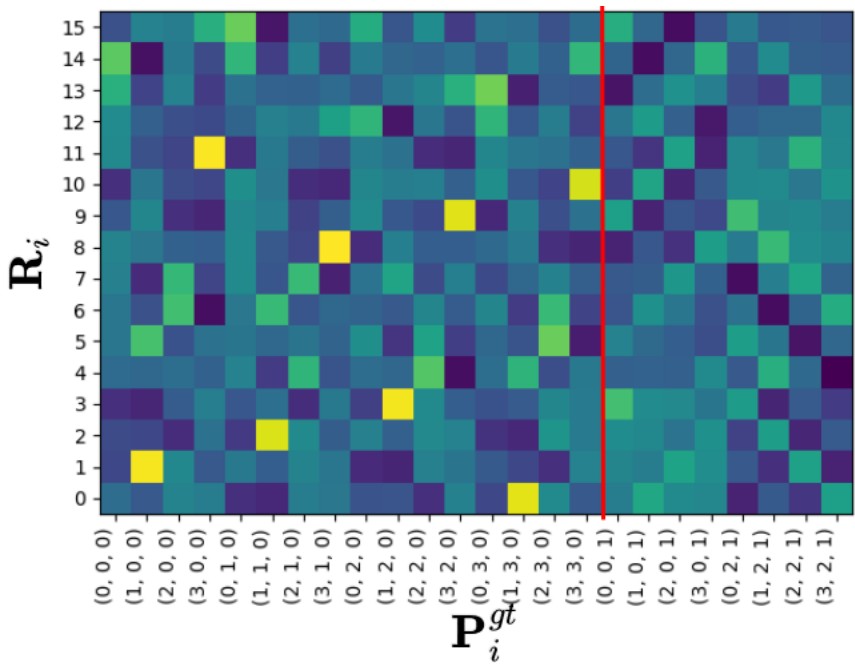

Figure 11: Cube dataset with $C_4 \times C_4$ rotations with $C_4 \times C_4 \times C_4$ base elements.

## B.5 Convolution matrix decomposition for Fig. 3.

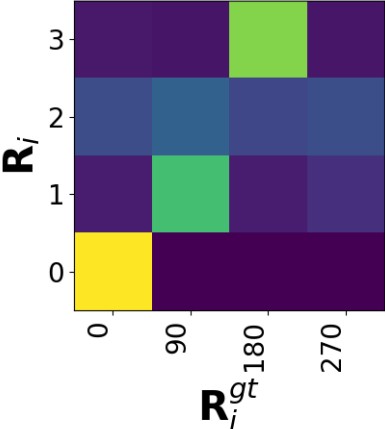

Figure 12: Coefficients for $C_4$ representations for $d = 25$ and learned representations of the lifting layer of the rotatedMNIST experiment.

## C Architectural details

In the current section, we outline some design choices used across all experiments. Code is available at https://github.com/computri/learnable-weight-sharing. Firstly, when defining the learnable representation stack $\mathbf{R}$, we have found that anchoring an identity element aids in distinguishing the base kernel from its potential transformations. This approach is based on the intuition of starting with a learnable base kernel and subsequently learning its transformations and we have found that it aids optimization.

### C.1 Regularizers

We test two regularizers on the representations $\mathbf{R}$, which are similar to those used by [38]:

- **Entropy Regularizer**: The primary motivation for using the entropy regularizer is to encourage sparsity in our weight-sharing schemes, which helps the matrices approximate actual permutation matrices rather than simply being doubly stochastic. This approach stems from the intuition for some transformations, the weight-sharing schemes should mimic soft-permutation matrices. The effectiveness of this sparsity depends on the specific group transformations relevant to the task—for example, C4 rotations are typically represented by exact permutation matrices. In contrast, $C_N$ rotations or scale transformations might require interpolation, thus aligning more closely with soft permutation matrices. Our experimental results indicate that the utility of this regularizer varies with the underlying transformations in the data; for instance, it is not beneficial for scale transformations in the MNIST dataset, as anticipated. The entropy regularizer is of the following form:

$$\text{ent}(\mathbf{R}) = - \sum_{ijk}^{N,D,D} \mathbf{R}_{ijk} \cdot \log(\mathbf{R}_{ijk})$$

- **Normalization Regularizer**: Empirically, we have found the normalization regularizer essential for reducing the number of iterations needed by the Sinkhorn operator to ensure the matrices are row and column-normalized. Without this regularizer, the tensors either fail to achieve double stochasticity or require an excessively high number of Sinkhorn iterations to do so. The normalization regularizer is of the following form:

$$\text{norm}(\mathbf{R}) = \frac{1}{D} \sum_{i}^{D} \text{sum}_r(\mathbf{R})_i^2 + \text{sum}_c(\mathbf{R})_i^2$$

Where $\text{sum}_r, \text{sum}_c$ are defined as in 4.2.

### C.2 MNIST

**Model architecture**  For all MNIST experiments, a simple 5-block CNN was used. Each block uses a kernel size of 5 and is succeeded by instance norm and ReLU activation, respectively. After the final convolution block, any spatial and group dimensions are reduced through a global average pooling operation, and a single linear layer is used as a classification head. For the group convolutional model and our weight sharing model, the default hidden channel dimension in the blocks was set to 32 unless otherwise stated, and 64 in the regular CNN models.

**Training details**  The models used a learning rate of 1e-2 and were trained for 100 epochs. All the experiments were done on a single GPU with 24GB memory under six hours.

### C.3 CIFAR10

**Model architecture**  We used the ResNet architecture as in [16] Appendix B.1, except that we swapped the final global max pooling operator with a global mean pooling. However, in contrast to [16], we use regular discrete kernels instead of continuous kernel parameterizations.

**Training details**  Following [16], we trained the models for 200 epochs using a learning rate of 1e-4. All the experiments were done on a single GPU with 24GB memory under six hours.

### C.4 Computational Demands

Fig. 13 14 show the computational scaling analysis of our weight sharing layer, comparing it to a group convolutional model of the same dimensions. We highlight that regular group convolutions can be implemented via weight-sharing schemes, resulting in equal computational demands for both approaches. Since the weight-sharing approach applies the group transformation in parallel across all elements (as a result of matrix multiplication and reshape operations), our method can prove quite efficient. Regarding memory allocation, group convolutions are often implemented using for-loops

over the group actions, and this sequential method imposes a less heavy memory burden since the operation is applied in series per group element. However, although scaling is quadratic w.r.t. the number of group elements and the domain size for weight-sharing, we mitigate this issue by use of the typically lower-dimensional support of convolutional filters (i.e., and ), rendering our approach practical for a wider range of settings.

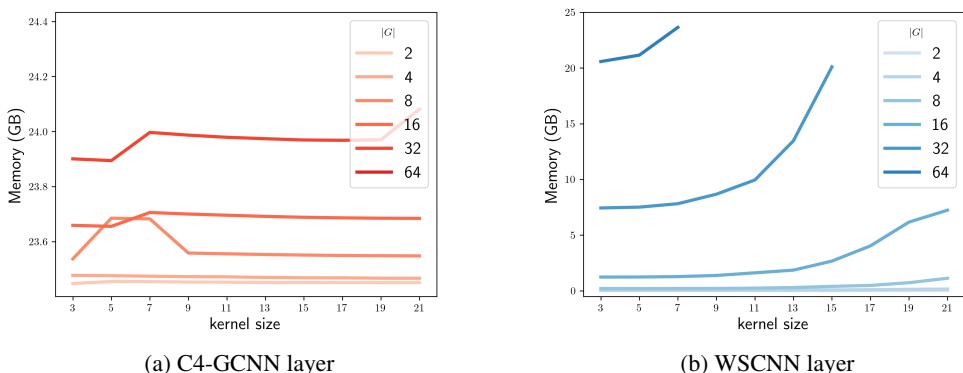

(a) C4-GCNN layer                    (b) WSCNN layer

Figure 13: Memory allocation at inference time for a C4-GCNN layer and a weight sharing layer, for different number of group elements $|G|$ and different kernel sizes. The input is $32 \times 3 \times 100 \times 100$ (batch $\times$ channels $\times$ height $\times$ width)

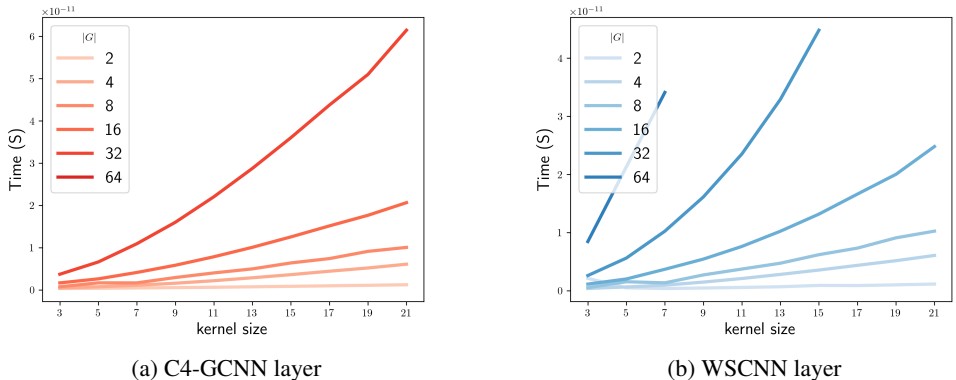

(a) C4-GCNN layer                    (b) WSCNN layer

Figure 14: Execution time at inference time for a C4-GCNN layer and a weight sharing layer, for different number of group elements $|G|$ and different kernel sizes. The input is $32 \times 3 \times 100 \times 100$ (batch$\times$ channels $\times$ height $\times$ width)

