# OpenReview forum: "Learning symmetries via weight-sharing with doubly stochastic tensors"
_NeurIPS.cc/2024/Conference — NeurIPS 2024 poster_

### Official Review · Reviewer_xU9K · 2024-07-10

**Soundness:** 3
**Presentation:** 3
**Contribution:** 2
**Rating:** 3
**Confidence:** 3

**Summary:**

In contrast to many works that impose strict architectural constraints to parameterize neural networks that are exactly equivariant to a known group, this work considers learning approximate equivariances to unknown groups. This is done using soft weight sharing schemes, where doubly stochastic tensors are learned and applied on canonical weights. The approach generalizes GCNNs, and is shown to learn natural symmetries in experiments.

**Strengths:**

1. Nice background section. In my opinion, it is written much better than other related papers in the area.
2. The method is a natural generalization of group convolutions, which is developed via the neat perspective of group convolutions as applications of transformations of a canonical weight tensor. It is nice that you can choose the number of "group elements", so that e.g. non-group-symmetries can be captured.

**Weaknesses:**

1. Section on "Regular representaitons allow for element-wise activations" is a little out of place and imcompletely justified. The claim that non-elementwise activations "in practice are not as effective as the classic element-wise activations" is a strong statement that needs more specific justification.
2. Direct parameterization of the kernels for each group element is expensive. Empirical runtime and memory analysis would be appreciated, to see the effect of this.
3. Empirical results are rather limited and weak. Few baselines are considered (see below as well), the baselines already perform well on these tasks, and WSCNN does not improve over the baselines in Table 1. Given that the method does not appear to be too scalable (see above), it is unclear where it would be useful in improving performance.
4. Other baselines (besides standard CNNs and GCNNs) and ablations are missing. For instance, one can imagine removing the sinkhorn projection onto the doubly stochastic matrices. If we initialize the $\Theta_i^l$ to be permutation matrices (your fixing of one representation to be identity is related to this), then this may also perform similarly.

**Questions:**

1. How do the learned filters of WSCNN look on CIFAR10?
2. Do learned representations ever look the same across different "group" elements?

**Limitations:**

Some discussion in section 6

---

> ### Author Rebuttal · Authors · 2024-08-07
>
> **Response:**
>
> Dear reviewer, we thank you for your time and valuable comments, which we address in the following.
>
> - On the use of element-wise activations: Regular group convolutions offer the benefit of having scalar feature field co-domains, permitting the use of any point-wise activation function without compromising equivariance [1, 2]. In contrast, steerable methods employ irreducible representations, which do not necessitate domain discretization [2]. However, these methods restrict the use of general point-wise activation functions, requiring specialized ones instead, which can be computationally expensive. This may additionally introduce discretization artifacts that disrupt exact equivariance, ultimately constraining the method’s expressivity [2].
> - Regarding computational analysis, we have included such an analysis comparing our weight-sharing layer to a conventional group convolution of comparable size in the additional results, please consider point 5 in the general rebuttal, and Fig. 2 and 3 in the attached PDF.
> - Regarding our empirical results and the usefulness of our method: we present additional experiments that demonstrate our method matches the performance of an unconstrained CNN with the same amount of effective but free kernels (128-channels) at a lower parameter budget (please also consider point 4 in the general rebuttal). In addition, we can outperform group convolutional methods for the discovery of partial and unknown symmetries. We also address this in points 1 and 2 of the general rebuttal, and Fig. 1a, 1b and 4 in the attached PDF. We hope this sufficiently addresses your concerns about experimental validation.
> - If we initialize the weight-sharing schemes as permutation matrices implementing a specific group transformation, we indeed recover a standard group convolution. A fair comparison of baselines would be comparing to methods that use weight-sharing for images, such as reference [31]. However, they report difficulties with end-to-end training and require meta-learning, making a direct comparison with our approach less straightforward. Following your comments, we have added the row-normalized baseline from [30] on rotated MNIST in the additional results (please consider point 3 in the general rebuttal, and Table 2 in the attached PDF). Visual inspection of the weight-sharing schemes seem to imply it is less effective in picking up meaningful group structures.
>
> Regarding your explicit questions:
>
> - (Question 1) Regarding the learned representations in the CIFAR-10 experiments, we highlight that our model operates *without* applying specific algebraic or group constraints to the learned representations, resulting in a more abstract and adaptable weight-sharing scheme. As a result, the patterns in weight-sharing are not as readily interpretable through traditional group-theoretic approaches, and it also makes direct visual interpretation of these patterns more nuanced.
> - (Question 2) While we do not put any explicit regularization on the diversity of the learned representations, we observed in our experiments that our method learns distinct weight-sharing patterns.
>
> We thank you for your insightful comments and would appreciate you considering raising your score if you believe they have been addressed sufficiently, or follow up with any questions we may help clarify further.
>
> **References**
>
> [1] General E(2)-Equivariant Steerable CNNs. Weiler et al. NeurIPS 2019
>
> [2] Fast, Expressive SE(n) Equivariant Networks through Weight-Sharing in Position-Orientation Space. Bekkers et al. ICLR 2024
>
> [30] Equivariance Discovery by Learned Parameter-Sharing. Raymond A. Yeh, Yuan-Ting Hu, Mark Hasegawa-Johnson, Alexander Schwing. AISTATS 2022
>
> [31] Meta-Learning Symmetries by Reparameterization. Allan Zhou, Tom Knowles and Chelsea Finn. ICLR 2021

---

> > ### Comment · Reviewer_xU9K · 2024-08-12
> >
> > Hello authors. Thank you for the response, and apologies for my late response.
> >
> > Your elaboration on (Question 1) is helpful (although it is clear from your paper, I forgot since I was thinking in terms of equivariant networks). I think that this, as well as your new ablations on less constraining of the learned representations, would be useful additions to your paper.
> >
> > My remaining worries are more high-level, on the utility of the method. In my view, the utility of approximate equivariance or discovering symmetries is not adequately demonstrated or achieved with the given method. The experiments are only on augmentations of MNIST, or on non-SOTA regimes in CIFAR10. This could be fine, if the model were developed with methods that one could see being useful in different regimes in the future. However, I think the poor efficiency and scalability of the model severely harms this. On the axes of efficiency and accuracy, the model is not efficient, and does not have clear accuracy gains in interesting experimental areas. Perhaps this work would benefit from expriements in areas (say, in the physical sciences) that more clearly desire equivariance, where equivariant models are actually SOTA or near SOTA.

---

> > > ### Author Response · Authors · 2024-08-12
> > >
> > > Thank you for your response, and we are glad some of your concerns have been addressed. Regarding your remaining concern about state-of-the-art performance, it is important to highlight that the primary aim of our approach is in line with regular group convolutional methods, focusing on learning useful constrained functions rather than competing directly with SOTA image models. This distinction is crucial as it frames our method as a tool for understanding and utilizing inherent symmetries in data. Note that the method can easily be extended to any task where signals act on a discretized domain, such as volumetric grids. In these cases there is some evidence that having unconstrained models are more prone to overfitting [1] and our method's ability to inherently recognize and utilize symmetries can be particularly advantageous. For the sake of developing the method, we consider them out of scope for this work.
> > >
> > > As such, in the current experiments we utilize MNIST and CIFAR-10 to demonstrate the main claims about symmetry discovery and feasibility. Additionally, while many CNNs rely on extensive data augmentation to achieve SOTA results, there is emerging evidence suggesting that such strategies might not always align well with real-world data distributions, potentially leading to suboptimal generalization [2, 3, 4, 5]. By contrast, our constrained approach offers a systematic way to learn and adapt to symmetries present in the data.
> > >
> > > In direct comparisons, our model shows that it can match the performance of unconstrained CNNs, underlining its efficacy even without clear accuracy gains in the standard experimental setups used (see point 4 in the general response). In these cases the CNN models have a considerably larger number of trainable parameters, implying that our method may not need to operate at the same scale/channel capacity to achieve satisfactory performance. This underlines the method's potential utility in settings where understanding and incorporating data-driven symmetries are crucial. We believe that further development and application of this method in contexts where equivariance is highly valued will substantiate its utility and is left as future work.
> > > We kindly encourage considering the broader context of our research’s objectives and its potential contributions to the field.
> > >
> > > **References**
> > >
> > > [1] Regular SE(3) Group Convolutions for Volumetric Medical Image Analysis. Thijs P. Kuipers and Erik J. Bekkers. MICCAI 2023
> > >
> > > [2] Learning Equivariances and Partial Equivariances from Data. David W Romero, Suhas Lohit. NeurIPS 2022
> > >
> > > [3] Learning Invariances in Neural Networks. Gregory Benton, Marc Finzi, Pavel Izmailov, Andrew Gordon Wilson. NeurIPS 2020
> > >
> > > [4] Learning Layer-wise Equivariances Automatically using Gradients. Tycho F. A. van der Ouderaa and Alexander Immer and Mark van der Wilk. 2023
> > >
> > > [5] Relaxing Equivariance Constraints with Non-stationary Continuous Filters. Tycho FA van der Ouderaa, David W Romero, Mark van der Wilk. NeurIPS 2022

---

> > > > ### Comment · Reviewer_xU9K · 2024-08-13
> > > >
> > > > We thank the authors for sharing their thoughts on the scope and context of the work within the literature. My worries still hold, so I maintain my score. GCNNs came out 8 years ago, and basic unconstrained CNNs on MNIST and CIFAR10 are quite far from empirical practice. In my opinion, experiments that actually significantly benefit from equivariance / approximate equivariance / equivariance detection would substantially improve the paper. There are many tasks / application areas that the community claims do or may benefit from equivariance, so it would be good to see experiments in such areas. I worry that factors such as the scalability of your method could harm its applicability in these tasks / applications, so experiments would be needed to reject this hypothesis.

---

> > > > > ### Author Response · Authors · 2024-08-14
> > > > >
> > > > > Thank you for your continued engagement and feedback on our submission. We appreciate your concerns and take them seriously as they help us refine our approach and clarify our contributions.
> > > > >
> > > > > We understand that the empirical practice has evolved since the introduction of GCNNs, among other models with known/built-in equivariance structures. However, the field of automatic (partial) symmetry discovery is fairly new, and the ability to detect and utilize equivariances automatically and the state of current research (i.e. [1, 2, 3, 4, 5]) operates in controlled and well-understood environments in order to validate the fundamental aspects of the proposed method.
> > > > >
> > > > > While we recognize the desire for experiments in application areas that could directly benefit from equivariance, our current focus is on establishing a strong theoretical and empirical foundation for our method. Such foundational work is essential for understanding the potential and limitations of new techniques before they can be successfully applied to more demanding domains.
> > > > >
> > > > > Your point on scalability and applicability in more practical scenarios is well-taken. We see this as an important direction for future research, where the scalability of our method can be further developed and tested in more complex environments. We plan to explore this in future work.
> > > > >
> > > > > **References**
> > > > >
> > > > > [1] Learning Equivariances and Partial Equivariances from Data. David W Romero, Suhas Lohit. NeurIPS 2022
> > > > >
> > > > > [2] Learning Invariances in Neural Networks. Gregory Benton, Marc Finzi, Pavel Izmailov, Andrew Gordon Wilson. NeurIPS 2020
> > > > >
> > > > > [3] Learning Layer-wise Equivariances Automatically using Gradients. Tycho F. A. van der Ouderaa, Alexander Immer, Mark van der Wilk. 2023
> > > > >
> > > > > [4] Relaxing Equivariance Constraints with Non-stationary Continuous Filters. Tycho FA van der Ouderaa, David W Romero, Mark van der Wilk. NeurIPS 2022
> > > > >
> > > > > [5] Bispectral Neural Networks, Sophia Sanborn, Christian Shewmake, Bruno Olshausen, Christopher Hillar. ICLR 2023

---

### Official Review · Reviewer_uLBU · 2024-07-11

**Soundness:** 3
**Presentation:** 2
**Contribution:** 2
**Rating:** 4
**Confidence:** 5

**Summary:**

Overview:
This paper claims to build upon a long line prior work on symmetry detection and learning equivariant kernels. [18,19,20,22, 25-31. Eespecially reference [30] which is an identical weight and parameter sharing scheme that learns  and discovers equivariances  solely by row-stochastic entries.  This paper's idea is to enforce both row and column stochasticity through  use of the Sinkhorn operator  to achieve equivariance discovery and applicable to  more interesting data domains such as images.

Advantages:
* The paper addresses an important problem of  Kernel equivariance via symmetry detection and weight  in CNN . However its marginal progress over prior work.

Weaknesses:
* Comparisons with prior work are missing.
* The quantitative advantage of  double stochasticity from row and column has advantages for metric spaces (images),. However this is not explained nor quantified through comparison.
* Moreover while ground truth tests are done for some toy problems, the quantitative generalizability of the double stochasticity over single stochasticity are never delineated, nor proved.

Questions:
This paper is identical to  this ICML 2024 workshop paper.   https://scholar.google.com/citations?view_op=view_citation&hl=en&user=gfRkDXEAAAAJ&citation_for_view=gfRkDXEAAAAJ:ufrVoPGSRksC ?
Also why no comparisons to even reference [30] ?

Missing References:
* Several references are missing, including the entire suite of probabilistic symmetry detection and equivariant NN is not discussed. See for e.g.   @misc{bloemreddy2020probabilisticsymmetriesinvariantneural,
      title={Probabilistic symmetries and invariant neural networks},
      author={Benjamin Bloem-Reddy and Yee Whye Teh},
      year={2020},
      eprint={1901.06082},
      archivePrefix={arXiv},
      primaryClass={stat.ML},
      url={https://arxiv.org/abs/1901.06082},
}

**Strengths:**

This paper enforces both row and column stochasticity through  use of the Sinkhorn operator  to achieve equivariance discovery.  However it's not clear what quantitative benefits this paper's method makes.

**Weaknesses:**

Weaknesses:
* Comparisons with prior work are missing.
* The quantitative advantage of  double stochasticity from row and column has advantages for metric spaces (images),. However this is not explained nor quantified through comparison.
* Moreover while ground truth tests are done for some toy problems, the quantitative generalizability of the double stochasticity over single stochasticity are never delineated, nor proved.

Questions:
This paper is identical to  this ICML 2024 workshop paper.   https://scholar.google.com/citations?view_op=view_citation&hl=en&user=gfRkDXEAAAAJ&citation_for_view=gfRkDXEAAAAJ:ufrVoPGSRksC ?
Also why no comparisons to even reference [30] ?

Missing References:
* Several references are missing, including the entire suite of probabilistic symmetry detection and equivariant NN is not discussed. See for e.g.   @misc{bloemreddy2020probabilisticsymmetriesinvariantneural,
      title={Probabilistic symmetries and invariant neural networks},
      author={Benjamin Bloem-Reddy and Yee Whye Teh},
      year={2020},
      eprint={1901.06082},
      archivePrefix={arXiv},
      primaryClass={stat.ML},
      url={https://arxiv.org/abs/1901.06082},
}

**Questions:**

Questions:
This paper is identical to  this ICML 2024 workshop paper.   https://scholar.google.com/citations?view_op=view_citation&hl=en&user=gfRkDXEAAAAJ&citation_for_view=gfRkDXEAAAAJ:ufrVoPGSRksC ? Is this ok ??

Also why no comparisons to even reference [30] ?

Missing References:
* Several references are missing, including the entire suite of probabilistic symmetry detection and equivariant NN is not discussed. See for e.g.   @misc{bloemreddy2020probabilisticsymmetriesinvariantneural,
      title={Probabilistic symmetries and invariant neural networks},
      author={Benjamin Bloem-Reddy and Yee Whye Teh},
      year={2020},
      eprint={1901.06082},
      archivePrefix={arXiv},
      primaryClass={stat.ML},
      url={https://arxiv.org/abs/1901.06082},
}

**Limitations:**

There are no attempts on analyzing the generalizability of the operator to varying initial and boundary conditions or Reynold numbers.

Other potentially relevant paper to consider:
https://www.sciencedirect.com/science/article/pii/S0021999123001997
This paper uses error-correction in neural operators based on the residual.

---

> ### Author Rebuttal · Authors · 2024-08-07
>
> Dear reviewer, we thank you for your time and comments, which we address in the following.
>
> Regarding the motivation for double stochasticity and its comparison to prior works, we point out that there are key benefits about employing double stochasticity over single stochasticity:
>
> - Theoretical motivation: We establish a link between regular representations which forms the basis of group convolution actions, and their manifestation as soft permutation matrices. Using double stochastic matrices allows for an analysis of learned representations through the lens of partial equivariance, a connection we elaborate on in Section 4.2. We are happy to motivate this in more detail in the camera-ready version.
> - Practical Implications: As outlined in Section 4.2, our methodology employs doubly stochastic matrices for weight-sharing, which facilitates direct calculation of expectations from the implicitly learned distribution. This setup avoids the need for Monte Carlo approximations and is adaptable to unrecognized symmetries that may not conform to standard group structures. This approach to handling partial equivariance offers a novel angle compared to traditional symmetry discovery methods, such as those proposed by Yeh et al. [30] and Zhou et al. [31].
>
> Hence, we propose that there is a well-grounded preference for employing double stochasticity over single stochasticity in the context of group convolutions. Following your comments, we have added the row-normalized baseline from [30] on rotated MNIST in the additional results (please consider point 3 in the general rebuttal, and Table 2 in the attached PDF).
>
> - As per author guidelines on dual submissions (section on Dual Submission, [**https://neurips.cc/Conferences/2024/CallForPapers**](https://neurips.cc/Conferences/2024/CallForPapers)) it states that *“Papers previously presented at workshops are permitted, so long as they did not appear in a conference proceedings (e.g., CVPR proceedings), a journal or a book.”* The work referenced in your review was submitted as an extended abstract at the ICML workshop “Geometry-grounded Representation Learning and Generative Modeling” (https://gram-workshop.github.io/cfp.html) which is **non-archival, and hence does not violate dual submission policy.**
> - Regarding the stated limitation, we appreciate the remark but do not focus on initial and boundary conditions as we do not use PDEs as a use case for our method in this work. If we were to extend our proposed method to PDE solving, we would require updating our architecture to accommodate boundary conditions and initial conditions to solve any PDEs efficiently. As far as the scope of this current work is concerned, we *learn* the symmetries or partial symmetries directly from the data through weight-sharing schemes, which do not require explicitly defining any boundaries.
>
> We thank you for your insightful comments and would appreciate you considering raising your score if you believe they have been addressed sufficiently, or follow up with any questions we may help clarify further.
>
> **References**
>
> [30] Equivariance Discovery by Learned Parameter-Sharing. Raymond A. Yeh, Yuan-Ting Hu, Mark Hasegawa-Johnson, Alexander Schwing. AISTATS 2022
>
> [31] Meta-Learning Symmetries by Reparameterization. Allan Zhou, Tom Knowles and Chelsea Finn. ICLR 2021

---

> ### Author Response · Authors · 2024-08-12
>
> Dear reviewer uLBU, we believe to have addressed any weaknesses raised in your original review, an acknowledgment or response to our rebuttal would be much appreciated. We are very much open to address any concerns in the remaining discussion period.

---

### Official Review · Reviewer_3Y5V · 2024-07-12

**Soundness:** 3
**Presentation:** 3
**Contribution:** 4
**Rating:** 7
**Confidence:** 4

**Summary:**

The paper proposes a symmetry discovery through learning parameter-sharing in weight matrices. The parameterization relies on relaxing underlying permutation matrices by transforming them as doubly stochastic matrices. In combination with additional regularization, the parameterization can be used to successfully discovery symmetries in data, improving generalization performance.

**Strengths:**

The method appears to be very elegant in offering a natural relaxation of underlying weight-sharing scheme through Sinkhorn operator. The paper is well-written and provides nice illustrations which guide the reader in understanding the proposed method. Then, the paper provides thorough empirical validation demonstrating usefulness of the approach in practice.

**Weaknesses:**

-Regularization.
The paper proposes a novel parameterization that can be used for symmetry discovery. In terms of objective, the work relies on direct regularization to avoid equivariants solutions, similar to some other symmetry discovery works. It has been shown in prior work that this strategy can have issues, since it introduces an additional hyperparameter which may need additional tuning (thereby is less ‘automatic’). In terms of explaining the methodology, the paper would benefit from some discussion on the role of the used regularizer.

-Scalability.
The paper seems to scale quadratically in |X|. Is this not an issue? How does this compare against scaling of alternative methods?

**Questions:**

-Regularization
For experiments, it would be helpful if it is clear how the strength of this regularizer is chosen. Cross-validation?
-Analysis of learned symmetries
It would be interesting to better understand what weight-sharing is being learned for CIFAR-10, apart from merely measuring improved test accuracy. Is there an analysis on the learned symmetry?
-Regularization
Entropy regularization seems to hurt on CIFAR-10, but improve on MNIST experiments. Do authors have an intuition on why this is the case?

**Limitations:**

The method proposes an elegant and novel symmetry discovery method. I expect the proposed parameterization to be beneficial to the community. The paper could improve a bit on the objective function side of things (seems to rely on directly encouraging symmetry, like some prior work). Apart from the remaining questions, the paper makes for a strong contribution.

---

> ### Author Rebuttal · Authors · 2024-08-07
>
> Dear reviewer, we thank you for your time and valuable comments, which we address in the following.
>
> Regarding employed regularizers, we acknowledge that the motivation could have been more explicitly mentioned, and we provide some added details:
>
> - Entropy Regularizer: The primary motivation for using an entropy regularizer is to encourage sparsity in our weight-sharing schemes, which helps the matrices approximate true permutation matrices rather than “simply” satisfying double stochasticity. This approach stems from our initial intuition that weight-sharing schemes should mimic soft permutation matrices. The effectiveness of the induced sparsity depends on the specific group transformations relevant to the task. For example, $C_4$ rotations are typically represented by exact permutation matrices. In contrast, $C_N$ rotations or scale transformations might require interpolation, thus aligning more closely with soft permutation matrices. Our experimental results indicate that the utility of this regularizer varies with the underlying transformations in the data. For instance, as anticipated it is of lesser use for scale transformations in the MNIST dataset.
> - Normalization Regularizer: Empirically, we have found the normalization regularizer essential for reducing the number of iterations needed by the Sinkhorn operator to ensure the matrices are row and column-normalized. Without this regularizer, the tensors either fail to achieve double stochasticity or require an excessively high number of Sinkhorn iterations to do so.
>
> We will outline our motivation behind these two regularizers and the impact of varying $\lambda$ values for the entropy regularizer more explicitly in the camera-ready version.
>
> - Regarding concerns about scalability: while we agree that quadratic scaling in $|X|$ can be impractical, potentially leading to polynomial $O(s^2)$ scaling in terms of image width $s$, our method exploits the fact that convolutional kernels have limited localized supports in practice. Thus, for a kernel size $k$ we scale with $|X| = k^2$ rather than the entire signal domain. This approach significantly reduces computational demands and makes our approach practically scalable. Please also consider point 5 in the general rebuttal for this aspect.
> - Regarding the interpretation of the weight-sharing patterns learned on CIFAR-10, it is important to note that our model does not impose algebraic or fixed group constraints on the learned representations. This design choice enhances the flexibility and abstraction of the weight-sharing schemes. As a result, the weight-sharing patterns may not be easily interpreted using conventional group-theoretic approaches, which could affect the direct visualization clarity of the learned weight-sharing.
>
> We thank you for your insightful comments and would appreciate you considering raising your score if you believe they have been addressed sufficiently, or follow up with any questions we may help clarify further.

---

> ### Author Response · Authors · 2024-08-12
>
> Dear reviewer 3Y5V, we believe to have addressed any weaknesses raised in your original review, an acknowledgment or response to our rebuttal would be much appreciated. We are very much open to address any concerns in the remaining discussion period.

---

> > ### Comment · Reviewer_3Y5V · 2024-08-12
> >
> > I appreciate the author's further explanations and comments. I remain my score of a 7 and recommend acceptance, as I deem this a technically solid paper that will have high impact in the community. This is conditional on author's including some more discussion on the use of regularization and the computational concerns, as well as the issues raised by other reviewers.

---

> > > ### Author Response · Authors · 2024-08-14
> > >
> > > We thank you for your response. We are glad you appreciated the work and will incorporate the necessary alterations.

---

### Official Review · Reviewer_1myY · 2024-07-21

**Soundness:** 2
**Presentation:** 2
**Contribution:** 3
**Rating:** 6
**Confidence:** 4

**Summary:**

This paper introduces a parameterisation that contains the ability to represent weight tying corresponding to arbitrary (?) group equivariances. In practice it can represent interpolations of weight tying, but it is argued that this is a feature (not a bug!) since strict equivariance is often too strong a constraint to place on a model for it to still fit the training data.

Strict equivariance is known to be represented by a permutation matrix (correct me if I'm wrong), which justifies the need to use doubly stochastic matrices to interpolate. This makes the edges of the parametisation strict equivariances. (For equivariances of discretised continuous signals this is not quite true, as the permutation in continuous space would need to be approximated by some interpolation in the discrete space)

This leads to a new parameterisation of a weight structure in a layer that can be trained in the usual way. The doubly-stochastic matrices can then be investigated to see if equivariance is actually learned.

The experiments implement the method, and run on benchmark datasets and synthetic datasets, showing good performance, and somewhat interpretable group structure appearing. It is unclear what the actual point of the experiments is, since there are many reasons to use equivariance, but the experiments are not phrased in terms of this (see discussion).

**Strengths:**

The problem of learning equivariances is very important, as it would remove a significant difficulty in designing networks with the correct inductive biases. The solution is flexible, as any (?) group structure can be represented by the parameterisation. There is also an elegant solution to the problem of needing a large number of parameters, that will work in practice for image data: Assuming translational equivariance, and only parameterising additional equivariances on the filters that are much smaller than the image.

**Weaknesses:**

Overall, the paper presents a well-reasoned method to an important problem, and I do believe that it meets the standard for publication at NeurIPS.

**Method & Presentation**
The method is well-justified. However, a final summary of what a forward pass through a layer looks like was not given, and would be really helpful. In addition, it would be helpful to have a clearer discussion of how many additional parameters are added (beyond lines 220 onwards), with the architectures that are discussed given as an explicit example.

Essentially, one thing which seems to be the case, but is not explicitly acknowledged, is that this method collapses to just a special parameterisation of weight matrices, where the weights have low-rank combined with doubly-stochastic structure. The low-rank-ness is shown in eq 6. While this is a simplistic way of looking at the method, it does give a helpful alternative view. Making this explicit would help the paper.

**What is the claim of the experiments?**
The experiments are the main weakness of the paper. Some qualitative results about the structure of the learned weights are given, which are helpful. But it is not clear what the quantitative claim of the experiments is. Equivariance can help in several ways, e.g. better out-of-distribution prediction, better prediction at low data, or smaller/compacter models. So is the claim that the equivariance inductive bias helps, and it can be discovered automatically? But in this case this is not disentangled from the model capacity. Perhaps a normal CNN would perform better if it were just made larger! This is additionally indicated that the baseline 70% accuracy on CIFAR 10 is low compared to what other non-group-equivariant methods can achieve. This unclarity also exists in the synthetic experiments, where the size of the dataset is not discussed. Low-data experiments could help here, since it's easier to make the model large enough that size doesn't help any longer, which isolates inductive bias only.

The CIFAR experiments show that making the model larger improves performance. How can we be sure that this is really the benefit of learning equivariances, rather than adding more capacity? Another experiment that is necessary here, is a comparison to a weight structure that does not have the doubly-stochastic constraint enforced on it. This would allow the effect of simply adding capacity to be tested.

Alternatively, a low-data experiment would allow the generalisation capabilities of the model to be tested (e.g. [Invariance Learning in Deep Neural Networks with Differentiable Laplace Approximations](fig 3 in https://proceedings.neurips.cc/paper_files/paper/2022/file/50d005f92a6c5c9646db4b761da676ba-Paper-Conference.pdf)). This could be done on MNIST variants, where currently the differences are so small, it is hard to draw conclusions.

This same issue pops up in the synthetic experiments. What is the dataset size? If the dataset is so large that all transformed signals are in the dataset, even a fully-connected network would learn the correct function. A low-data experiment is needed to really show that truly an equivariance has been learned that can help to _generalise_. Alternatively, you need to argue a benefit on the basis of parameter count.

In summary: The claims that the experiment section support are not clear. The field of equivariances is mature enough that the potential benefits have been clearly described, and these need to be clearly evaluated in experimental sections.

**Related Work**
The idea of relaxing equivariance by placing a distribution over transformations is older than the papers currently cited. E.g.:
- [Local Group Invariant Representations via Orbit Embeddings](http://proceedings.mlr.press/v54/raj17a/raj17a.pdf)

The discussion of "symmetry discovery methods" is unclear. What data do these methods need, or what kind of training signal do they use? What kind of predictive improvements do they obtain? Is the goal of these papers the same as those in the previous paragraph? Or, is the way that these methods learn group structure different from those in the previous section? If so, how?

Methods that learn a degree of equivariance on a layer-by-layer basis are relatively new, and it would be good to discuss this explicitly. E.g.:
- [Residual Pathway Priors for Soft Equivariance Constraints](https://proceedings.neurips.cc/paper/2021/file/fc394e9935fbd62c8aedc372464e1965-Paper.pdf) Finzi et al is not mentioned at all, but does allow partial equivariance in a layer-by-layer way.
- Reference [26] "Learning layer-wise equivariances automatically using gradients" is cited, but only in the context of arguing that overly constrained models suffer poor performance, but not in the context that this paper also discusses how to learn the right equivariance to use. This author has more papers on learning invariance/equivariances that may be relevant.

Overall, the literature review misses a lot of relevant work. The suggestions I gave are off the top of my head and I definitely missed some important papers too. However, it is the responsibility of the authors to put the time and effort into going beyond this to give a more thorough overview.

**Minor**
- _"Requiring no prior knowledge of the possible symmetries."_ (line 52) It is true that earlier methods (with the exception of [31]) could only pick between groups that were completely specified a-priori. While this paper _in principle_ does provide a parameterisation that can _search_ over a much wider space, this space needs to be limited for scalability reasons, and it was not demonstrated that the method would work reliably without this "prior" being added!
- It would be really helpful to have a full discussion of the impact on the number of parameters, for these specific experiments.Needs a discussion of the total number of parameters
- Is there a typo in params in table 1? Under "Params" should "103 + 265K" be "103K + 256K"?

**Questions:**

- Can this parameterisation represent arbitrary group equivariances? Is the representation of every strict equivariance a permutation matrix? It would be helpful to be explicit about how general this really is.
- Am I right in understanding that this method ultimately just parameterises a low-rank weight matrix over different feature channels?
- Can the benefit that equivariance promises in the low-data regime still be provided by this method when the invariance is learned? Since effectively, you're just parameterising weights in a different way (low rank?). In rotationally equivariant settings in low-data, could these weights not just overfit, rather than learning to rotate filters? Would this not lose an important benefit of equivariance?
- How important is the doubly-stochastic nature of things? Could you just run an experiment without the Sinkhorn component at all?
- Can you give a very short (ideally 1 sentence, or a 2-3 sentences) summary of the quantitative claims that are made about the method, that are verified in the experiment section?

**Limitations:**

See above.

Overall, this is a really interesting idea. I do have some concerns about the evaluation. These concerns are large in the scheme of determining how well this method really works relative to clearly formulated claims, but small relative to typical approaches in the ML community.

---

> ### Author Rebuttal · Authors · 2024-08-07
>
> **Response:**
> Dear reviewer, we thank you for your constructive feedback and engaged questions, which we address in the following.
>
> - We thank the reviewer for pointing out the connection of our proposed method to low-rank approaches. It is possible to make our doubly stochastic matrices converge to permutation matrices with an appropriate regularization technique. In those cases we obtain sparse matrices that could be represented via low-rank approximation methods. However, we note that the degree of regularization is task- and domain-specific.
> - Regarding experimental claims, our claim is indeed that the inductive bias introduced by equivariance is beneficial, and can furthermore be discovered automatically [Ref Fig 1 a,b and Fig 4 in the PDF] For natural images, the bias of translation equivariance may be sufficient in many cases, in particular when enough channels are present. This is not the case for other datasets, where the data itself may contain group symmetries beyond translation, as shown by [20, 27]. Our main claim is the ability to uncover such symmetry groups and employ them in different downstream tasks. While a look-up table may match the existence of an *a priori* specified symmetry group, this is not the case for partial symmetries or incomplete ones. In such cases, we demonstrate that we can discover the symmetry in the sense of a useable shared representation without requiring to point out the exact subgroup (if such a subgroup even exists).
> - We acknowledge that conventional CNNs with an equivalent channel count inherently possess a higher expressive capacity due to a lack of kernel constraints. Therefore, we have opted to compare overall parameter counts rather than the number of kernels, a more comprehensive assessment in our opinion. For completeness, we have additionally measured the performance for an unconstrained 128-channel CNN (please consider point 4 in the general rebuttal).
> - We thank the reviewer for the suggestion on experimenting with the low-data regime. We agree that the size of the dataset plays an important role, and we have added the suggested experiments for both cases of full and partial symmetry discovery (please consider point 2 in the general rebuttal and Fig. 2 in the attached PDF).
> - Regarding additional literature review, we will expand the related works section and further clarify the position of our work within the literature for the camera-ready version. We would like to stress that we do *not* explicitly impose any group structure in our representation stack, since that would defeat our goal of symmetry *learning* and enables us to potentially learn a weight sharing scheme that violates some of the group axioms if the group symmetry is not present in the data. Therefore, we refrain from making any theoretical assumptions that make use of more complex statistical group theory results. We agree that we could have been more precise regarding these claims in the paper, and will clarify this. Our method primarily aims to learn symmetries from data explicitly through a weight sharing scheme which *may or may not* be an exact group symmetric structure (if the data does not possess a full group structure). Then, we implicitly assign several probability measures of finite support over this learned symmetry scheme. We emphasize that our method is therefore more flexible than a strict group equivariant network due to its ability to learn partial symmetries.
>
> Regarding your explicit questions:
>
> - (Question 1) Yes, this parameterization can represent arbitrary group equivariances for compact Lie groups. For strict or partial equivariance, we employ a permutation matrix.
> - (Question 2) We hope that our first point in the response above addressed this question.
> - (Question 3) In the low-data regime, the model might overfit to a limited set of group transformations. This can be mitigated with a proper regularizer. We would argue that this is an advantage for learning partial or incomplete symmetries, as we make no assumptions on completeness or uniform distribution over group transformation in the data. In those cases, strict equivariance can hurt performance and may also be an overconfident bias lacking the data evidence to support it. On the other hand, our method allows us to address this issue by learning directly from any symmetries that are present in the data, without the need of such assumptions on, e.g., completeness of a group structure or uniform distribution over group transformation.
> - (Question 4) Zhou et al. [31] essentially learn a parameter sharing scheme without constraints on the sharing pattern except norm regularization. However, they report that they require meta-learning to pick up on symmetries, while our method does not. We presume that the added constraint on double stochasticity is beneficial in that regard.
> - (Question 5) To briefly summarize in two points: (1) The model can recover group convolutional structures when clear symmetries, including partial symmetries, are present in the data (see our experiments on Rotated MNIST); and (2) The model can pick up on useful weight sharing patterns when there are no *a priori* known symmetry structures. This is shown in our CIFAR-10 experiments, where we match the CNN’s model performance and outperform a model with pre-specified group equivariance. This is additionally underlined in Point 1, 2 and 4 in the general response.
>
> We thank you for your insightful comments and would appreciate you considering raising your score if you believe they have been addressed sufficiently, or follow up with any questions we may help clarify further.
>
> **References**
>
> [20] Learning equivariances and partial equivariances from data. David W. Romero and Suhas Lohit. NeurIPS 2022
>
> [27] Approximately equivariant networks for imperfectly symmetric dynamics. Rui Wang, Robin Walters, Rose Yu. ICML 2022
>
> [31] Meta-Learning Symmetries by Reparameterization. Allan Zhou, Tom Knowles and Chelsea Finn. ICLR 2021

---

> ### Author Response · Authors · 2024-08-12
>
> Dear reviewer 1myY, we believe to have addressed any weaknesses raised in your original review, an acknowledgment or response to our rebuttal would be much appreciated. We are very much open to address any concerns in the remaining discussion period.

---

> ### Comment · Reviewer_1myY · 2024-08-14
>
> > We acknowledge that conventional CNNs with an equivalent channel count inherently possess a higher expressive capacity due to a lack of kernel constraints. Therefore, we have opted to compare overall parameter counts rather than the number of kernels, a more comprehensive assessment in our opinion. For completeness, we have additionally measured the performance for an unconstrained 128-channel CNN (please consider point 4 in the general rebuttal).
>
> This is a really helpful addition. Interesting, so the main claim now is that for a parameter-constrained model, you obtain better performance? Size of the model clearly does come into this, as the CNN also gets better as more filters are added. This is fine, and a clear claim. However this does mean that the current experiments only show that the invariance learning properties work in the _underfitting_ regime! This is different from what is wanted from learning equivariances, where we may want to show the ability to obtain improved performance, once other simpler methods like making the model larger stop working.
>
> This is why other methods e.g. [31, 26] consider second order information (meta-learning, marginal likelihood approximations): to distinguish between inductive biases even when the training loss cannot. This was also discussed in [*] and back in 2018 [**]. This is what the low data experiment could have shown, if it was verified that the model was large enough to sufficiently fit the training data. Also the low-data experiment is of limited usefulness, since it does not contain a baseline of a non-invariant model. The interesting question here is to see how much of an improvement you can get by learning invariances over a model that cannot do this. (Also relevant to your answer to my question 4.)
>
> > We would like to stress that we do not explicitly impose any group structure in our representation stack
>
> This was clear from the paper, and is certainly valuable! However, I don't see how this makes the literature I pointed to any less relevant?
>
>
> ## Overall
> Either way, while I don't think this is the end of the question of how to learn equivariance automatically, I do believe it is an interesting paper, and I will continue to argue for acceptance.
>
> [*] [Invariance Learning in Deep Neural Networks with Differentiable Laplace Approximations](https://proceedings.neurips.cc/paper_files/paper/2022/file/50d005f92a6c5c9646db4b761da676ba-Paper-Conference.pdf) 2022 (not cited in the paper)
>
> [**] [Learning Invariances using the Marginal Likelihood](https://arxiv.org/abs/1808.05563) 2018 (not cited in the paper)

---

### Author Rebuttal · Authors · 2024-08-07

# General response

We are grateful for the time, effort and invaluable feedback provided by the reviewers. We next address general points raised jointly amongst reviewers, and proceed to respond to specific comments in the individual author rebuttals below.

Firstly, we are glad that you **found the work interesting and elegant** (1myY: "an elegant solution to the problem”; “Overall, this is a really interesting idea”; 3Y5V: “The method appears to be very elegant in offering a natural relaxation of underlying weight-sharing scheme”). Furthermore, reviewers appreciated that the **theory was presented in a clear manner** (3Y5V: ”The paper is well-written”; xU9K: ”Nice background section. In my opinion, it is written much better than other related papers in the area.”).

The reviewers also identify valuable points for improvement which we will gladly incorporate via minor edits in the camera-ready version of the paper. The main additions and experiments in response to reviewers’ concerns include:

1. We have added two experiments that show the methods effectiveness in the case of partial and misspecified symmetries, namely: MNIST with partial group structure (rotations sampled from a subset of $SO(2)$ (**Fig. 1a** in the attached PDF) and CIFAR10 with horizontal flips.
2. We have added a comparison of models in the low-data regime for fully rotated and partially augmented MNIST (**Fig. 4** in the attached PDF). In the presence of partial symmetries (half of the rotation angles) and symmetry misspecification (flips) our weight sharing method can find useful weightsharing patterns.
3. Reviewers have pointed out a comparison to reference [30] and the benefits of employing double stochasticity over single stochasticity. We have attached results for this baseline on the rotatedMNIST dataset trained for 100 epochs. Visual inspection of the weight sharing schemes suggest the method is not as effective in picking up the group structure.
4. We have incorporated results for CIFAR-10 for a standard CNN model that matches the effective kernel size used in our GCNN/WSCNN models, namely 128 channels (calculated as $|G| * \text{channels} = 4*32 = 128$) (**Table 1** in the attached PDF). We have also added this model as a baseline in the flipped CIFAR-10 experiment (**Fig. 1b** in the attached PDF). Despite the kernel constraints in our model, it achieves performance comparable to that of the unconstrained 128-channel CNN, but with significantly less learnable parameters (our 2.1 M vs. the 6.5 M of the CNN).
5. We have added a computational scaling analysis of our weight sharing layer, comparing it to a group convolutional model of the same dimensions (**Fig. 2 and 3** in the attached PDF). We highlight that regular group convolutions can be implemented via weight-sharing schemes, resulting in equal computational demands for both approaches. Since the weight-sharing approach applies the group transformation in parallel across all elements (as a result of matrix multiplication and reshape operations), our method can prove quite efficient. Regarding memory allocation, group convolutions are often implemented using *for-loops* over the group actions, and this sequential method imposes a less heavy memory burden since the operation is applied in series per group element. However, although scaling is quadratic w.r.t. the number of group elements and the domain size for weight-sharing, we mitigate this issue by use of the typically lower-dimensional support of convolutional filters (i.e., $|G| <= 16$ and $\text{kernel size} <= 7$), rendering our approach practical for a wider range of settings.
6. We thank the reviewers (1myY, uLBU) for pointing out relevant related literature and will update our related works section accordingly with your suggested references.

We hope that our additional experiments and clarifications were able to resolve any commonly raised points, and would be happy to respond to remaining concerns, if any, during the discussion period.

We further address each reviewers' concerns and questions in our individual responses, and thank once again for the valuable feedback that we believe will increase the quality of our contributions.

**References**

[30] Equivariance Discovery by Learned Parameter-Sharing. Raymond A. Yeh, Yuan-Ting Hu, Mark Hasegawa-Johnson, Alexander Schwing. Proceedings of The 25th International Conference on Artificial Intelligence and Statistics, PMLR 151:1527-1545, 2022.

[31] Meta-Learning Symmetries by Reparameterization. Allan Zhou, Tom Knowles and Chelsea Finn. ICLR 2021

---

### Comment · Area_Chair_ijCz · 2024-08-11
**Reviewer-Author discussion period**

Dear Reviewers,

The deadline for the reviewer-author discussion period is approaching. If you haven't done so already, please review the rebuttal and provide your response at your earliest convenience.

Best wishes, AC

---

### Decision · Program_Chairs · 2024-09-25

**Decision:**

Accept (poster)

**Comment:**

This paper received mixed reviews. The reviewers generally acknowledge the technical contribution, considering the proposed method an interesting solution to an important problem. However, the negative evaluations focus primarily on the experiments: Reviewer 1myY noted that the current experiments are not the most effective for demonstrating the strength of the proposed method, and Reviewer xU9K raised concerns about the scalability and practicality of the approach. While these concerns are valid, I still believe the technical contribution of the paper meets the standard for acceptance.